# Tackling View-Dependent Semantics in 3D Language Gaussian Splatting

Jiazhong Cen [1] [*]  Xudong Zhou [1]  Jiemin Fang [2]  Changsong Wen [1]  Lingxi Xie [2]  Xiaopeng Zhang [2]  Wei Shen [1]  Qi Tian [2]

## Abstract

Recent advancements in 3D Gaussian Splatting (3D-GS) enable high-quality 3D scene reconstruction from RGB images. Many studies extend this paradigm for language-driven open-vocabulary scene understanding. However, most of them simply project 2D semantic features onto 3D Gaussians and overlook a fundamental gap between 2D and 3D understanding: a 3D object may exhibit various semantics from different viewpoints—a phenomenon we term **view-dependent semantics**. To address this challenge, we propose **LaGa** (Language Gaussians), which establishes cross-view semantic connections by decomposing the 3D scene into objects. Then, it constructs view-aggregated semantic representations by clustering semantic descriptors and reweighting them based on multi-view semantics. Extensive experiments demonstrate that LaGa effectively captures key information from view-dependent semantics, enabling a more comprehensive understanding of 3D scenes. Notably, under the same settings, LaGa achieves a significant improvement of **+18.7% mIoU** over the previous SOTA on the LERF-OVS dataset. Our code is available at: https://github.com/https://github.com/SJTU-DeepVisionLab/LaGa.

## 1. Introduction

3D Gaussian Splatting (3D-GS) (Kerbl et al., 2023), a recent breakthrough in radiance fields, offers superior rendering efficiency and quality. Its unstructured 3D point-cloud-like representation makes it a promising foundation for open-

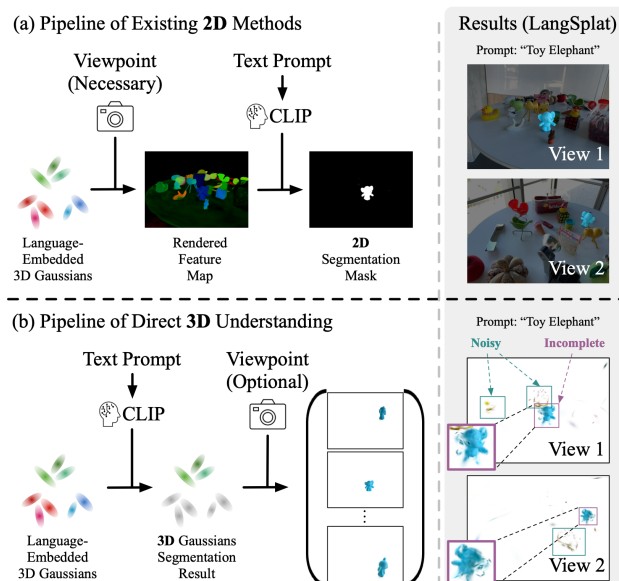

*Figure 1.* Pipeline comparison of existing 2D methods (a) and the direct 3D scene understanding paradigm (b). While 2D methods excel in pixel-wise understanding via rendered feature maps, they fail unexpectedly when 3D Gaussians are directly retrieved by matching learned 3D features with CLIP text embeddings.

vocabulary scene understanding. To this end, recent studies (Qin et al., 2024; Shi et al., 2024; Bhalgat et al., 2024; Peng et al., 2025; Cheng et al., 2024; Qu et al., 2024) extend 3D-GS by lifting multi-view 2D semantic features extracted from vision-language models like CLIP (Radford et al., 2021) into the 3D feature representation. As shown in Figure 1(a), these methods heavily rely on the differentiable rasterization mechanism of 3D-GS. During training, they optimize the 3D features by aligning rendered feature maps with 2D semantic features at corresponding viewpoints. At inference, they continue rendering the learned 3D semantic features into 2D feature maps, using them for pixel-wise semantic understanding with cross-view consistency.

However, when applying their learned 3D features to direct 3D perception (Wu et al., 2024b; Lee et al., 2025), as shown in Figure 1(b), their effectiveness degrades significantly, limiting their applicability to downstream tasks such

---

[*]Work done during internship at Huawei  [1]MoE Key Lab of Artificial Intelligence, AI Institute, Shanghai Jiao Tong University [2]Huawei Technologies Co., Ltd.. Correspondence to: Wei Shen <wei.shen@sjtu.edu.cn>, Jiemin Fang <jaminfong@gmail.com>.

*Proceedings of the 42nd International Conference on Machine Learning*, Vancouver, Canada. PMLR 267, 2025. Copyright 2025 by the author(s).

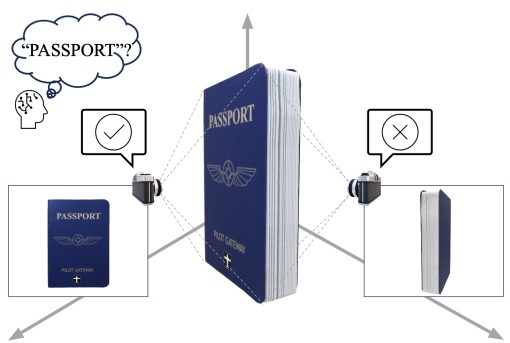

Figure 2. An illustration of view-dependent 3D semantics. The passport exhibits different semantics from different viewpoints.

as 3D object editing, AI-driven interaction, and precise 3D localization. We identify a fundamental issue behind this limitation: the **view-dependency of 3D semantics**. As illustrated in Figure 2, a passport viewed from the front clearly reveals its title, while from the back or side, it becomes unrecognizable. This phenomenon highlights a fundamental gap between 2D and 3D understanding. Simply projecting 2D semantics onto 3D Gaussians results in incomplete or inaccurate semantic assignments, as each Gaussian inherits semantics visible only from specific viewpoints. Specifically, this leads to false positives (noisy Gaussians) and false negatives (incomplete results), as depicted in Figure 1(b).

To quantify this issue, we conduct two analyses. First, a **semantic similarity distribution** analysis shows that multi-view semantic features of the same object often exhibit lower intra-object similarities than inter-object similarities, providing direct evidence of view-dependent semantics. Second, a **semantic retrieval integrity** analysis finds that about **50%** of 2D semantic features fail to retrieve their corresponding 3D objects completely, further validating its negative affect.

To address this challenge, we propose **LaGa**, a simple yet effective method for open-vocabulary 3D scene understanding with 3D-GS. LaGa first performs 3D scene decomposition by grouping multi-view 2D masks into coherent 3D objects, establishing cross-view semantic connections to explicitly capture view-dependent 3D semantics. Then, LaGa constructs view-aggregated semantic representations by extracting a representative set of semantic descriptors for each 3D object via adaptive clustering. To enhance robustness, LaGa assigns weights to descriptors based on two factors: (1) Global alignment, measuring the directional similarity between the descriptor and its global feature, *i.e.*, the average semantic embedding across all related 2D semantics; and (2) Internal compactness, reflecting the consistency of semantics within the descriptor's feature cluster. During inference, LaGa selects the highest weighted response of descriptors as the object's final output, thereby preserving

critical information across viewpoints.

Despite its simplicity, LaGa significantly bridges the performance gap between 3D and 2D perception on existing benchmarks, achieving **+18.7% mIoU** over previous 3D methods and surpassing state-of-the-art 2D results by **+8.8% mIoU**. Our approach offers a new perspective for advancing 3D semantic understanding within the 3D-GS framework.

## 2. Related Work

### 2.1. Radiance Fields and 3D Gaussian Splatting

Neural Radiance Fields (NeRF) (Mildenhall et al., 2020) pioneer using differentiable rendering for 3D scene reconstruction from multi-view images. Numerous works enhance its rendering quality (Zhang et al., 2020; Martin-Brualla et al., 2021; Barron et al., 2021; 2022; 2023) and efficiency (Sun et al., 2022; Chen et al., 2022; Lindell et al., 2021; Hedman et al., 2024; Reiser et al., 2021; Müller et al., 2022; Wizadwongsa et al., 2021; Neff et al., 2021; Fridovich-Keil et al., 2022). Recently, 3D Gaussian Splatting (3D-GS) (Kerbl et al., 2023) uses explicit 3D Gaussians as 3D representation and differentiable rasterization for real-time, high-quality rendering. It inspires applications in 3D generation (Yi et al., 2024; Yang et al., 2024a; Tang et al., 2025), 3D scene editing (Chen et al., 2024; Wang et al., 2024), and 4D scene reconstruction (Yang et al., 2024b; Wu et al., 2024a; Duan et al., 2024; Mihajlovic et al., 2025). We build upon 3D-GS to achieve open-vocabulary 3D scene understanding.

### 2.2. 3D Scene Understanding in Radiance Fields

**Semantic-Agnostic Scene Understanding.** The growing popularity of radiance fields leads to research in interactive 3D segmentation and scene decomposition. NVOS (Ren et al., 2022) introduces the first interactive method for object selection in NeRFs. N3F (Tschernezki et al., 2022), DFF (Kobayashi et al., 2022), and ISRF (Goel et al., 2023) lift features from 2D self-supervised models (Caron et al., 2021) into 3D via learned feature fields. NeRF-SOS (Fan et al., 2023) and ContrastiveLift (Bhalgat et al., 2023) distill 2D feature similarities into 3D. More recently, with the advent of the Segment Anything Model (SAM) (Kirillov et al., 2023), studies such as Garfield (Kim et al., 2024), OmniSeg3D (Ying et al., 2024), SA3D (Cen et al., 2023), GaGa (Lyu et al., 2024), and SAGA (Cen et al., 2025a) uses SAM-extracted segmentation masks for scene decomposition. Our approach builds upon this framework by using SAM-extracted masks to construct cross-view semantic connections and decompose the 3D scene into objects.

**Semantic-Aware Scene Understanding.** Prior to open-vocabulary methods, research primarily focus on extending 2D closed-set perception models to radiance fields. Semantic-NeRF (Zhi et al., 2021) explore semantic propa-

gation in NeRFs, while methods such as ObSuRF (Stelzner et al., 2021), DM-NeRF (Bing et al., 2023), Panoptic-NeRF (Fu et al., 2022), PCF-Lift (Zhu et al., 2024), NESF (Vora et al., 2022), and Siddiqui et al. (2023) employ 2D vision models for instance or panoptic segmentation in 3D. Additionally, Instance-NeRF (Liu et al., 2023b) and NeRF-RPN (Hu et al., 2023) introduce end-to-end models for instance detection within radiance fields. However, these methods are inherently constrained to closed-set categories.

Recent studies incorporate vision-language models for open-vocabulary scene understanding. LERF (Kerr et al., 2023) learns a feature field that mimics CLIP (Radford et al., 2021) features across multiple views, while 3D-OVS (Liu et al., 2023a) performs weakly-supervised segmentation in NeRFs. With the emergence of 3D-GS, methods such as LangSplat (Qin et al., 2024), LEGaussians (Shi et al., 2024), GOI (Qu et al., 2024), and N2F2 (Bhalgat et al., 2024) focus on efficiently encoding high-dimensional language features within the explicit 3D Gaussians using feature compression techniques like codebooks or hyperplane-based grid decomposition. However, these approaches rely on 2D rendered feature maps for perception, and show degraded performance in 3D scene understanding.

To address this limitation, OpenGaussian (Wu et al., 2024b) and Open3DRF (Lee et al., 2025) constrain 3D features directly instead of supervising 2D rendered features. Open-Gaussian employs scene decomposition for 3D point-wise perception but primarily enforces semantic consistency within objects. However, it overlooks view-dependent semantic variations and rigidly assigns semantics, leading to significant information loss and limiting its ability to achieve comprehensive 3D understanding. In contrast, we leverage scene decomposition to establish cross-view semantic connections, effectively capturing and preserving view-dependent semantics, thereby bridging the gap between direct 3D understanding and 2D pixel-wise understanding.

## 3. Preliminary

**3D Gaussian Splatting (3D-GS)** is a recent advancement in radiance fields that represents a 3D scene using a set of colored 3D Gaussians, $\mathcal{G} = \{\mathbf{g}_1, \mathbf{g}_2, \ldots, \mathbf{g}_N\}$, where $N$ is the number of Gaussians in the scene. Each Gaussian's mean defines its position, and its covariance determines its scale. By "splatting" these 3D Gaussians onto an image plane, 3D-GS enables real-time rendering.

Given a viewpoint with camera intrinsics and extrinsics, 3D-GS first projects the 3D Gaussians onto a 2D plane. The color $\mathbf{C}(\mathbf{p})$ of a pixel $\mathbf{p}$ is computed via alpha blending over an ordered set of Gaussians $\mathcal{G}_\mathbf{p}$ overlapping the pixel. Let $\mathbf{g}_i^\mathbf{P}$ denote the $i$-th Gaussian in $\mathcal{G}_\mathbf{p}$, the pixel color is:

$$\mathbf{C}(\mathbf{p}) = \sum_{i=1}^{|\mathcal{G}_\mathbf{p}|} \mathbf{c}_{\mathbf{g}_i^\mathbf{P}} \alpha_{\mathbf{g}_i^\mathbf{P}} \prod_{j=1}^{i-1}(1 - \alpha_{\mathbf{g}_j^\mathbf{P}}), \tag{1}$$

where $\mathbf{c}_{\mathbf{g}^\mathbf{P}}$ is the color of $\mathbf{g}_i^\mathbf{P}$, and $\alpha_{\mathbf{g}_i^\mathbf{P}}$ is computed from the corresponding 2D Gaussian with covariance $\Sigma$, scaled by a learnable per-Gaussian opacity.

To extend 3D-GS for scene understanding, many studies propose augmenting 3D Gaussians with additional attributes. A common approach is to attach a feature vector $\mathbf{f_g}$ to each Gaussian $\mathbf{g}$. This allows rendering a feature map using the same alpha blending formulation as in Equation (1):

$$\mathbf{F}(\mathbf{p}) = \sum_{i=1}^{|\mathcal{G}_\mathbf{p}|} \mathbf{f}_{\mathbf{g}_i^\mathbf{P}} \alpha_{\mathbf{g}^\mathbf{P}} \prod_{j=1}^{i-1}(1 - \alpha_{\mathbf{g}_j^\mathbf{P}}). \tag{2}$$

Specifically, language-embedded 3D Gaussian Splatting methods (Qin et al., 2024; Zhou et al., 2024; Bhalgat et al., 2024; Qu et al., 2024; Shi et al., 2024) align the rendered feature map $\mathbf{F}$ with CLIP's visual features during training. At inference, open-vocabulary understanding is performed by computing the relevance between a text query and the rendered feature map. However, these learned features are optimized for 2D perception. When directly applied to 3D understanding using the learned 3D features $\{\mathbf{f_g} \mid \mathbf{g} \in \mathcal{G}\}$, performance degrades significantly, as shown in Figure 1.

## 4. View-Dependency of 3D Semantics

As discussed in Section 1, a 3D object's semantics vary with viewpoint shifts. To quantitatively analyze this effect, we conduct two experiments. The data preprocessing for them is introduced in Section 5.2, which delivers a set of 2D masks and their corresponding semantic features. Using these masks, the 3D-GS scene is decomposed into structurally meaningful but semantic-agnostic 3D objects Section 5.3, establishing connections between 2D masks and their corresponding 3D Gaussians. We assume the retrieved 3D Gaussians represent a 'complete' 3D object.

We then introduce two analytical experiments: (1) Semantic similarity distribution analysis and (2) Semantic retrieval integrity analysis.

**Semantic Similarity Distribution.** After the 3D decomposition phase, the multi-view 2D masks along with the corresponding semantics of the same 3D object are clustered together. Thus, we can compute the cosine similarities of semantic features of the same object (intra-object) and between different objects (inter-object). This analysis is conducted across four scenes from the LERF (Kerr et al., 2023) dataset. The results are shown in Figure 3, where the two distributions demonstrate a significant overlap, indicating

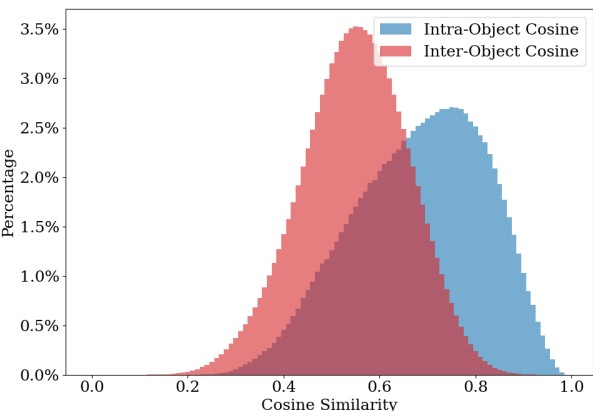

Figure 3. Distribution of intra- and inter-object cosine similarities.

that the multi-view semantics of a single 3D object often exhibit lower cosine similarities than those of semantics from different 3D objects. This statistical finding provides strong evidence of the existence of view-dependent semantics.

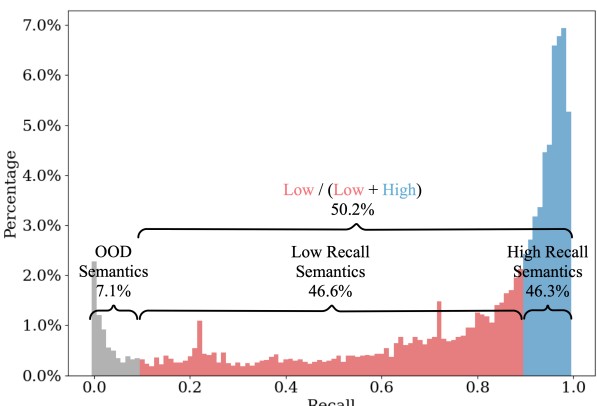

Figure 4. Distribution of recall rates for 2D semantic features.

**Semantic Retrieval Integrity.** We project multi-view 2D semantics onto 3D Gaussians and use them to retrieve 3D Gaussians from the scene (implementation details in Appendix B.3). Since each 2D mask corresponds to a 'complete' 3D object via scene decomposition, the recall rate—proportion of semantically retrieved 3D Gaussians within the corresponding region—evaluates how well the semantics capture the object. We define detected Gaussians as those with a cosine similarity above 0.75 to a given 2D feature. Results in Figure 4 show that low-recall features ([0.1, 0.9]) account for **50%**. Notably, for complex scenes containing numerous objects with 360° viewpoints, such as 'Figurines', this percentage rises to **61.9%**. This indicates

that a single object's Gaussians may exhibit varying 2D semantics, highlighting the need for multi-view semantic aggregation in 3D understanding.

## 5. Method

In this section, we introduce the overall pipeline of LaGa (Section 5.1) and its key components, including 3D scene decomposition (Section 5.3) and view-aggregated semantic representation (Section 5.4).

### 5.1. Overall Pipeline

Figure 5 presents the overall pipeline of LaGa. First, LaGa performs 3D scene decomposition to establish cross-view semantic connections by grouping multi-view 2D semantics corresponding to the same 3D object. This process naturally partitions the scene into 3D objects, which serve as fundamental carriers of multi-view semantics.

Next, LaGa constructs view-aggregated semantic representations to effectively preserve view-dependent semantics. It first adaptively extracts representative sets of semantic descriptors for 3D objects, consolidating multi-view semantics into a more holistic representation. Then, a weighted descriptor relevance aggregation strategy refines the importance of each descriptor, enhancing noise tolerance. Using these weighted descriptors, LaGa enables direct 3D scene understanding without relying on 2D rendered feature maps.

### 5.2. Data Preparation

We follow the data preparation pipeline introduced by Qin et al. (2024) to obtain 2D segmentation and semantics priors. Concrete extraction method is introduced in Appendix B.1. After the data preparation, we have:

- A set of 2D masks $\mathcal{M}^{\mathbf{I}} = \{\mathbf{M}_i^{\mathbf{I}} \in \{0, 1\}^{HW} \mid i = 1, \ldots, N_{\mathbf{I}}\}$ for each image $\mathbf{I}$ in the training set $\mathcal{I}$.

- A $l_2$-normalized feature $\mathbf{v}^{\mathbf{M}} \in \mathbb{R}^C$ corresponding to each mask $\mathbf{M}$, where $C$ denotes the feature dimension.

### 5.3. 3D Scene Decomposition

To address the view-dependency of 3D semantics, we aim to establish connections among different 2D semantics depicting the same 3D object. Since 2D semantic features are linked to multi-view 2D masks, and these masks inherently correspond to coherent 3D objects, these connections can be effectively constructed by decomposing the 3D scene.

Inspired by existing 3D-GS decomposition methods (Kim et al., 2024; Ying et al., 2024; Lyu et al., 2024; Cen et al., 2025a), we adopt a contrastive learning approach to train a set of Gaussian affinity features $\mathcal{F} = \{\mathbf{f_g} \in \mathbb{R}^{C'} \mid \mathbf{g} \in \mathcal{G}\}$, where $C'$ denotes the affinity feature dimension. These

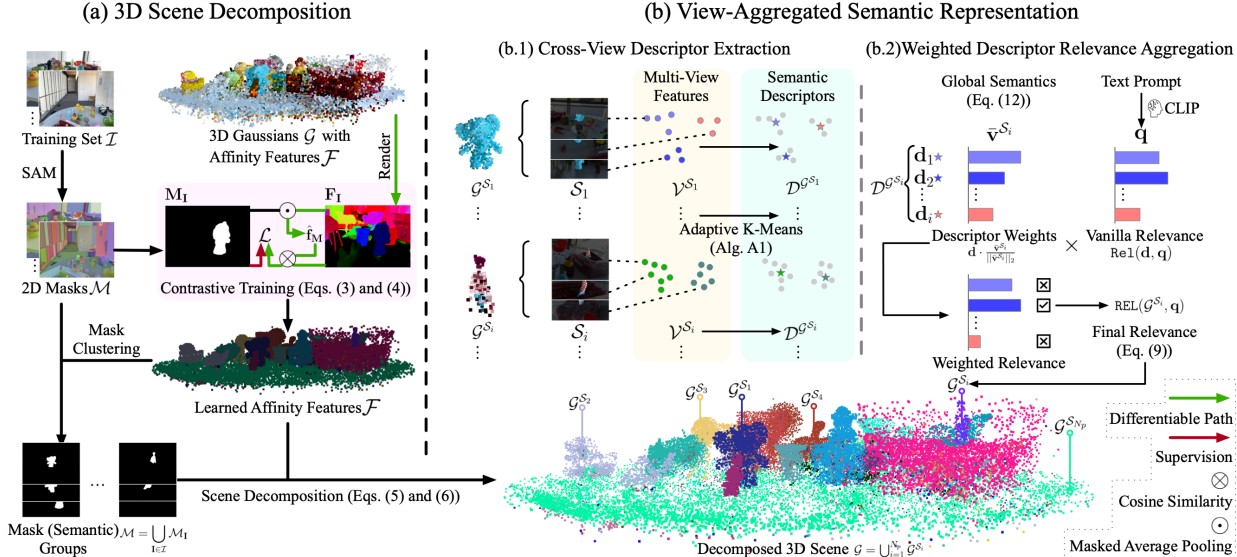

*Figure 5.* Overall pipeline of **LaGa**. LaGa first establishes cross-view semantic connections through contrastive 3D scene decomposition and then constructs view-aggregated semantic representations by adaptively clustering semantic descriptors and reweighting them.

features serve as indicators of whether two 3D Gaussians belong to the same object, ensuring that their similarity reflects their structural and spatial coherence.

To train these affinity features, we render a feature map $\mathbf{F_I}$ for each view $\mathbf{I}$ using Equation (2). We then assign a mask affinity prototype $\hat{\mathbf{f}}_{\mathbf{M^I}} \in \mathbb{R}^{C'}$ to each 2D mask $\mathbf{M^I}$ through masked average pooling:

$$\mathrm{MAP}(\mathbf{M^I}, \mathbf{F_I}) = \frac{1}{\sum_{\mathbf{p} \in \delta(\mathbf{I})} \mathbf{M^I}(\mathbf{p})} \sum_{\mathbf{p} \in \delta(\mathbf{I})} \mathbf{M^I}(\mathbf{p}) \mathbf{F_I}(\mathbf{p}). \tag{3}$$

Here, $\delta(\mathbf{I})$ denotes the set of pixels in image $\mathbf{I}$. The training objective encourages features within the same mask to cluster while separating those outside the mask:

$$\mathcal{L} = \sum_{\mathbf{I} \in \mathcal{I}} \sum_{\mathbf{M} \in \mathcal{M^I}} \sum_{\mathbf{p} \in \delta(\mathbf{I})} (1 - 2\mathbf{M}(\mathbf{p})) \max(\langle \hat{\mathbf{f}}_{\mathbf{M}}, \mathbf{F_I}(\mathbf{p}) \rangle, 0). \tag{4}$$

Though the loss is computed in 2D, after training, the affinity features and mask prototypes of the same 3D object will converge into a compact cluster, as the rendered features originate from the same group of 3D Gaussians.

Let $\mathcal{M} = \bigcup_{\mathbf{I} \in \mathcal{I}} \mathcal{M^I}$ be the set of masks of all training images. After training, we employ HDBSCAN (McInnes et al., 2017) to automatically cluster the masks in $\mathcal{M}$ based on their prototypes $\hat{\mathbf{f}}_{\mathbf{M}}$, resulting in $\mathcal{M} = \bigcup_{i=1}^{N_p} \mathcal{S}_i$, where $\mathcal{S}_i$ is a set of 2D masks that belong to the same 3D object and $N_p$ denotes the number of clusters. This step explicitly establishes cross-view connections among 2D masks, grouping multi-view semantics into coherent 3D objects.

Beyond connecting masks across views, mask prototypes also serve as references for identifying the 3D Gaussians associated with their corresponding object. The prototype of each 3D object is computed as:

$$\mathbf{t}^{\mathcal{S}_i} = \frac{1}{|\mathcal{S}_i|} \sum_{\mathbf{M} \in \mathcal{S}_i} \hat{\mathbf{f}}_{\mathbf{M}}. \tag{5}$$

The object index $i^*$ to which a Gaussian $\mathbf{g}$ is assigned is determined by maximizing the similarity between its affinity feature $\mathbf{f_g}$ and the prototype $\mathbf{t}^{\mathcal{S}_i}$ of each object $\mathcal{S}_i$:

$$i^* = \arg\max_i \langle \mathbf{f_g}, \mathbf{t}^{\mathcal{S}_i} \rangle. \tag{6}$$

where $\langle \cdot, \cdot \rangle$ represents the cosine similarity.

At this stage, the scene is decomposed into $N_p$ distinct objects, represented as $\mathcal{G} = \bigcup_{i=1}^{N_p} \mathcal{G}^{\mathcal{S}_i}$. Each object $\mathcal{G}^{\mathcal{S}_i}$ corresponds to a group of 2D masks $\mathcal{S}_i$, effectively preserving multi-view semantics across the scene. Moreover, these objects enable the 3D Gaussians within each object to share a common set of semantics, significantly reducing storage requirements and accelerating inference. In the next section, we detail the process of extracting semantic descriptors for these objects based on their related semantics.

**Why Is 3D Scene Decomposition Unaffected by View-Dependency?** Unlike high-level semantics, which need to encode rich semantic information, 2D segmentation primarily captures object boundaries and thus remains stable across viewpoints. Except in extreme cases, SAM reliably produces accurate segmentations, allowing LaGa to establish robust multi-view semantic connections.

## 5.4. View-Aggregated Semantic Representation

For each object $\mathcal{G}^{\mathcal{S}_i}$ with multi-view 2D masks $\mathcal{S}_i$, we denote its multi-view semantic features as:

$$\mathcal{V}^{\mathcal{S}_i} = \{\mathbf{v}^{\mathbf{M}} \mid \mathbf{M} \in \mathcal{S}_i\}. \tag{7}$$

LaGa obtains a robust 3D view-aggregated semantic representation from these multi-view 2D semantics through two key steps: 1) Cross-view descriptor extraction: Generates an informative set of representative semantic descriptors; 2) Weighted descriptor relevance aggregation: Adjusts the importance of each descriptor to mitigate noise.

**Cross-View Descriptor Extraction.** After the 3D scene decomposition, $\mathcal{V}^{\mathcal{S}_i}$ maintain all available multi-view semantic information about the 3D object $\mathcal{G}^{\mathcal{S}_i}$, it is crucial to extract a representative set of descriptors that effectively summarize the semantic variations observed across different viewpoints. To achieve this, we apply K-means clustering to $\mathcal{V}^{\mathcal{S}_i}$, obtaining a set of cluster centroids:

$$\mathcal{D}^{\mathcal{G}^{\mathcal{S}_i}} = \left\{\mathbf{d}_i \in \mathbb{R}^{C'} \mid i \in \{1, \dots, N^{\mathcal{G}^{\mathcal{S}_i}}\}\right\}. \tag{8}$$

These centroids serve as **semantic descriptors** for the object $\mathcal{G}^{\mathcal{S}_i}$. Since semantic complexity varies across objects, we determine the number of descriptors $N^{\mathcal{G}^{\mathcal{S}_i}}$ adaptively using the silhouette score (see Algorithm 1 for details).

**Weighted Descriptor Relevance Aggregation.** Instead of treating all descriptors equally, we introduce a weighting mechanism that prioritizes more reliable descriptors. Given a text query $\mathbf{q} \in \mathbb{R}^C$ encoded by CLIP, we define the object-level relevance score as:

$$\text{REL}(\mathcal{G}^{\mathcal{S}_i}, \mathbf{q}) = \max_{\mathbf{d} \in \mathcal{D}^{\mathcal{G}^{\mathcal{S}_i}}} \omega^{\mathbf{d}} \cdot \text{Rel}(\mathbf{d}, \mathbf{q}). \tag{9}$$

The relevance score $\text{Rel}(\mathbf{d}, \mathbf{q})$ follows Kerr et al. (2023):

$$\text{Rel}(\mathbf{d}, \mathbf{q}) = \min_i \frac{\exp\langle \mathbf{d}, \mathbf{q} \rangle}{\exp\langle \mathbf{d}, \mathbf{q} \rangle + \exp\langle \mathbf{d} \cdot \phi^i_{\text{canon}} \rangle}, \tag{10}$$

where $\phi^i_{\text{canon}}$ represents canonical text embeddings[1].

The weight $\omega^{\mathbf{d}}$ for each descriptor $\mathbf{d}$ is based on two criteria:

$$\begin{aligned} \omega^{\mathbf{d}} &= \mathbf{d} \cdot \frac{\bar{\mathbf{v}}^{\mathcal{S}_i}}{||\bar{\mathbf{v}}^{\mathcal{S}_i}||_2} \\ &= \underbrace{\frac{\mathbf{d}}{||\mathbf{d}||_2} \cdot \frac{\bar{\mathbf{v}}^{\mathcal{S}_i}}{||\bar{\mathbf{v}}^{\mathcal{S}_i}||_2}}_{\text{(i) Directional Consistency}} \times \underbrace{||\mathbf{d}||_2}_{\text{(ii) Internal Compactness}}, \end{aligned} \tag{11}$$

$$\bar{\mathbf{v}}^{\mathcal{S}_i} = \frac{1}{|\mathcal{V}^{\mathcal{S}_i}|} \sum_{\mathbf{v} \in \mathcal{V}^{\mathcal{S}_i}} \mathbf{v}, \tag{12}$$

---

[1]Following prior work, the canonical phrases are "object," "thing," "texture," and "stuff".

- **Directional Consistency** measures the cosine similarity between each descriptor and the global feature $\bar{\mathbf{v}}^{\mathcal{S}_i}$ of its corresponding object. Descriptors that aligns with the dominant semantics of the object are assigned higher weights. For instance, the spine of a book may appear as a "knife" from certain viewpoints, which is suppressed due to semantic inconsistency. In contrast, descriptor resembling a "passport" is more semantically aligned with the global semantics "book," and is thus assigned higher weights.

- **Internal Compactness** quantifies intra-cluster semantic agreement via the L2 norm of a descriptor. Semantically consistent descriptors have compact clusters and yield higher norms. In contrast, if the features of a descriptor are inconsistent with diverse directions, their vector average will cancel out, resulting in a lower norm. Thus, the norm serves as a confidence measure for semantic reliability.

Together, these criteria allow LaGa to suppress unreliable descriptors. Since the view-aggregated semantic representations are assigned to 3D objects, LaGa enables direct 3D understanding without relying on rendered feature maps.

# 6. Experiments

In this section, we first introduce the datasets and evaluation protocol. We then present both quantitative and qualitative results to demonstrate the effectiveness of LaGa. Additionally, we conduct extensive ablation studies to analyze the impact of different design choices in LaGa.

## 6.1. Datasets and Evaluation Protocol

We evaluate LaGa on **LERF-OVS** (Kerr et al., 2023; Qin et al., 2024), **3D-OVS** (Liu et al., 2023a), and **ScanNet** (Dai et al., 2017). LERF-OVS consists of complex 360° indoor scenes, while 3D-OVS features forward-facing scenes with long-tailed categories. Both datasets provide 2D annotations. Unlike prior methods that perform segmentation on rendered 2D feature maps, we conduct 3D segmentation at the Gaussian level, generating binary Gaussian segmentation maps ('1' for foreground, '0' for background). These maps are then rendered into 2D views and evaluated against ground-truth masks.

For ScanNet, we follow OpenGaussian (Wu et al., 2024b) to perform 3D point cloud semantic segmentation, directly comparing predictions with point-wise ground-truth annotations for a more direct assessment of 3D perception quality.

## 6.2. Quantitative Results

**LERF-OVS Dataset.** Table 1 compares LaGa with both 3D and 2D approaches, demonstrating substantial improve-

*Table 1.* Quantitative mIoU (%) results on the LERF-OVS dataset: "F." (Figurines), "T." (Teatime), "R." (Ramen), and "W." (Waldo Kitchen). '†' indicates results from Wu et al. (2024b), '‡' indicates our reimplementation, and '*' denotes unrefereed preprints.

|   | METHODS | F. | T. | R. | W. | MEAN |
|---|---|---|---|---|---|---|
| 2D | LSEG | 7.6 | 21.7 | 7.0 | 29.9 | 16.6 |
|  | LERF | 38.6 | 45.0 | 28.2 | 37.9 | 37.4 |
|  | LEGAUSSIANS | 60.3 | 44.5 | 52.6 | 41.4 | 46.9 |
|  | LANGSPLAT | 44.7 | 65.1 | 51.2 | 44.5 | 51.4 |
|  | N2F2 | 47.0 | 69.2 | 56.6 | 47.9 | 54.4 |
|  | OCCAMLGS* | 58.6 | 70.2 | 51.0 | 65.3 | 61.3 |
|  | VLGS* | 58.1 | 73.5 | 61.4 | 54.8 | 62.0 |
| 3D | OPENGAUSSIAN† | 39.3 | 60.4 | 31.0 | 22.7 | 38.4 |
|  | SAGA‡ | 36.2 | 19.3 | 53.1 | 14.4 | 30.7 |
|  | LANGSPLAT‡ | 25.9 | 35.6 | 29.3 | 33.5 | 31.1 |
|  | LEGAUSSIANS‡ | 31.2 | 34.5 | 17.6 | 17.3 | 25.2 |
|  | OPENGAUSSIAN‡ | 61.1 | 59.1 | 29.2 | 31.9 | 45.3 |
|  | SUPERGSEG* | 43.7 | 55.3 | 18.1 | 26.7 | 35.9 |
|  | LAGA (OURS) | **64.1** | **70.9** | **55.6** | **65.6** | **64.0** |

*Table 2.* Quantitative mIoU (%) results on the 3D-OVS dataset. Methods marked with '*' denote concurrent preprints.

| METHODS | BED | BENCH | ROOM | SOFA | LAWN | MEAN |
|---|---|---|---|---|---|---|
| LERF | 73.5 | 53.2 | 46.6 | 27.0 | 73.7 | 54.8 |
| 3D-OVS | 89.5 | 89.3 | 92.8 | 74.0 | 88.2 | 86.8 |
| GOI | 89.4 | 92.8 | 91.3 | 85.6 | 94.1 | 90.6 |
| LEGAUSSIANS | 84.9 | 91.1 | 86.0 | 87.8 | 92.5 | 88.5 |
| LANGSPLAT | 92.5 | 94.2 | 94.1 | 90.0 | 96.1 | 93.4 |
| N2F2 | 93.8 | 92.6 | 93.5 | 92.1 | 96.3 | 93.9 |
| OCCAMLGS* | 96.8 | 95.8 | 96.5 | 88.8 | 97.0 | 95.0 |
| VLGS* | 96.8 | 97.3 | 97.7 | 95.5 | 97.9 | 97.1 |
| LAGA (OURS) | 96.8 | 92.8 | 97.0 | 93.0 | 96.9 | 95.3 |

*Table 3.* Quantitative results on the ScanNet dataset.

| METHODS | 19 CLASSES mIoU / mACC. | 15 CLASSES mIoU / mACC. | 10 CLASSES mIoU / mACC. |
|---|---|---|---|
| LEGAUSSIANS | 3.8 / 10.9 | 9.0 / 22.2 | 12.8 / 28.6 |
| LANGSPLAT | 3.8 / 9.1 | 5.4 / 13.2 | 8.4 / 22.1 |
| OPENGAUSSIAN | 24.7 / 41.5 | 30.1 / 48.3 | 38.3 / 55.2 |
| LAGA (OURS) | **32.5 / 49.1** | **35.5 / 53.5** | **42.6 / 63.2** |

ments over previous state-of-the-art methods. Unlike 2D methods that rely on rendered feature maps, LaGa performs direct 3D segmentation at the Gaussian level. Under identical settings, LaGa achieves an **18.7%** mIoU improvement over OpenGaussian (Wu et al., 2024b)[2].

Notably, LaGa achieves significant gains in the "Waldo Kitchen" scene, where OpenGaussian's rigid feature assignment fails to capture key distinguishing semantics that appear inconsistently across viewpoints. This result highlights the importance of LaGa's view-aggregated semantic representation in addressing view-dependent semantics.

Furthermore, LaGa outperforms recent preprints, including OccamLGS (Cheng et al., 2024), VLGS (Peng et al., 2025), and SuperGSeg (Liang et al., 2024), establishing a new benchmark for open-vocabulary 3D segmentation.

**3D-OVS Dataset.** As shown in Table 2, LaGa achieves competitive 3D segmentation performance on the 3D-OVS dataset, reaching a 95.3% mIoU, comparable to state-of-the-art 2D methods. The lack of significant improvement can be attributed to the dataset's lower complexity—each scene contains only a few distinctive objects, and existing methods already achieve near-optimal performance (over 90% mIoU). Additionally, the forward-facing scene configuration and limited viewpoint variations reduce the impact of view-dependent semantics, which LaGa is designed to handle. These results align with our expectations.

**ScanNet Dataset.** The detailed implementation details of point cloud segmentation can be found in Appendix B.6.

The results are shown in Table 3. Predictably, both 2D segmentation methods, LangSplat (Qin et al., 2024) and LEGaussians (Shi et al., 2024), fail to produce reasonable results due to poor 3D point feature quality. In contrast, LaGa outperforms OpenGaussian, demonstrating superior capability in handling point cloud segmentation.

**Difference between LaGa and OpenGaussian.** Both OpenGaussian (Wu et al., 2024b) and LaGa decompose 3D scenes for open-vocabulary understanding. However, OpenGaussian assigns a 2D CLIP feature to each Gaussian via a rule-based representative view selection, which overlooks the information in multi-view semantics and results in suboptimal performance. In contrast, LaGa extracts semantic descriptors through adaptive multi-view clustering and applies weighted relevance aggregation to suppress noise and enhance robustness. This design enables LaGa to achieve a significantly more robust 3D semantic representation, with an improvement of +18.7% in mIoU.

## 6.3. Qualitative Results

We provide a visual comparison with existing methods in Figure 6. Compared to 2D methods such as LEGaussians (Shi et al., 2024) and LangSplat (Qin et al., 2024), LaGa generates sharper boundaries and more complete 3D segmentations. While these prior methods can coarsely detect objects, they suffer from multiple false positives and false negatives due to the inherent limitation of learning 3D Gaussian features from restricted viewpoints.

For example, in the "Waldo Kitchen-Ottolenghi" example,

---

[2]We revise the evaluation script of OpenGaussian's official code. As a result, our reimplementation achieves +6.9% mIoU higher than originally reported.

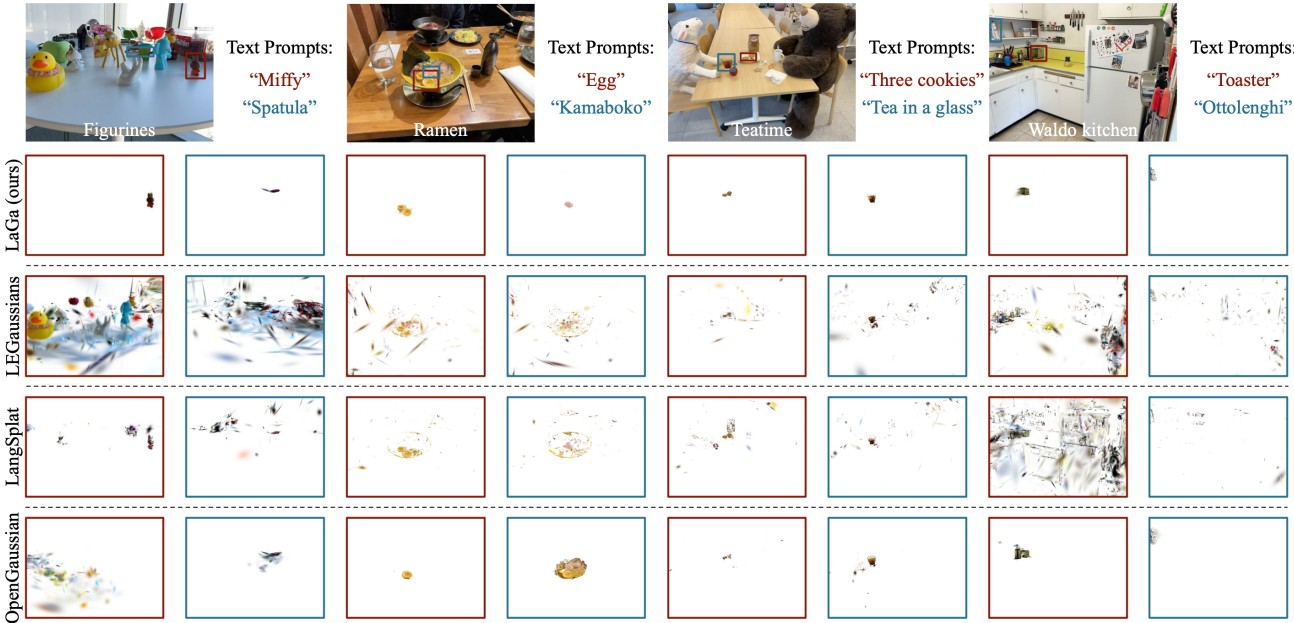

*Figure 6.* Qualitative comparison on the LERF-OVS dataset. LaGa delivers more precise 3D segmentations within 3D-GS. Although corresponding views are provided with colored bounding boxes for clarity, neither viewpoint information nor visual prompts are used.

both LEGaussians and LangSplat capture only a partial outline of the book, as CLIP can only recognize its title from certain perspectives. In most views, the title is unidentifiable, leading to incomplete feature fusion and missing regions. In contrast, LaGa effectively captures the book's full semantics by integrating information across multiple views.

OpenGaussian (Wu et al., 2024b) also produces clean and distinct segmentations by enforcing feature consistency within pre-segmented 3D regions. However, it fails to account for view-dependent semantics, as it rigidly assigns a single 2D feature to an entire 3D region. This limitation results in frequent misclassifications (*e.g.*, Figurines-Miffy, Ramen-Egg, and Waldo Kitchen-Toaster). These findings further underscore the necessity of preserving multi-view semantics and highlight the effectiveness of LaGa.

To further demonstrate the generalizability of LaGa, we present visualization results on the MIP-360 dataset (Barron et al., 2022). As illustrated in Figure 7, LaGa performs well in both complex indoor and outdoor environments. For more qualitative results, including more visual comparisons, multi-view and multi-granularity segmentation results, visualizations on the 3D-OVS dataset, and examples of mask clustering and scene editing, see Appendix C.

### 6.4. Ablation Study

Our ablation studies evaluate the effectiveness of the view-aggregated semantic representation, focusing on cross-view descriptor extraction and weighted descriptor relevance ag-

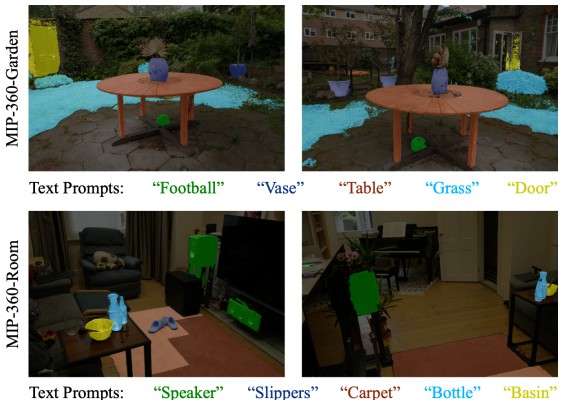

*Figure 7.* Visualization results on the MIP-360 dataset. LaGa shows accurate localization across diverse textual prompts in both indoor and outdoor complex environments.

gregation. To assess the impact of cross-view descriptor extraction, we introduce two simple baseline methods for deriving semantic descriptors of 3D objects:

1. **Average Pooling**: Represent the object $\mathcal{G}^{\mathcal{S}_i}$ by computing the average of the visual features in $\mathcal{V}^{\mathcal{S}_i}$.

2. **Max Pooling**: Retain all features in $\mathcal{V}^{\mathcal{S}_i}$ and, during inference, use the highest response among them as the representative response for the object $\mathcal{G}^{\mathcal{S}_i}$.

The results are presented in Table 4. "Fixed: $k$" denotes clustering the multi-view features using a fixed number of

*Table 4.* Ablation study on the LERF-OVS dataset. For each scene, the best and second-best results are highlighted in **bold** and underlined fonts. "DW" represents the descriptor weighting scheme introduced in Section 5.4. Within each "DW" experiment group, superior results are marked with a gray background.

| Methods | | F. | T. | R. | W. | Mean |
|---|---|---|---|---|---|---|
| Avg. Pooling | | 48.6 | 64.9 | 47.8 | 52.4 | 53.4 |
| Max Pooling | | 38.4 | 42.7 | 35.4 | 56.7 | 43.3 |
| Fixed: 5 | – DW | **64.7** | 58.9 | 50.5 | 62.2 | 59.1 |
| | + DW | 63.2 | 64.5 | **56.2** | 63.5 | 61.9 |
| Fixed: 10 | – DW | 62.8 | 68.0 | 46.3 | 60.3 | 59.4 |
| | + DW | 59.5 | **71.3** | 54.3 | 63.9 | 62.3 |
| Fixed: 20 | – DW | 61.1 | 63.5 | 41.6 | 63.0 | 57.3 |
| | + DW | 58.2 | 63.2 | 48.9 | **66.4** | 59.2 |
| Adaptive | – DW | 59.7 | 65.5 | 53.8 | 62.8 | 60.4 |
| | + DW$^c$ | 61.6 | 69.3 | 53.6 | 59.2 | 60.9 |
| | + DW$^d$ | 60.8 | 67.3 | 55.6 | 63.9 | 61.9 |
| | + DW | 64.1 | 70.9 | 55.6 | 65.6 | **64.0** |

Bag-of-Words Effect in CLIP:

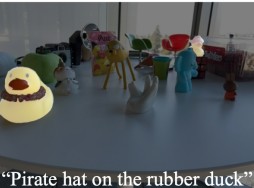 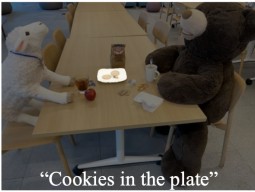

"Pirate hat on the rubber duck"    "Cookies in the plate"

Lack of Context in 2D Semantics:

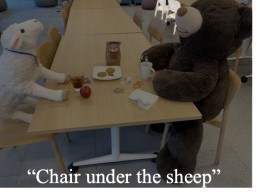 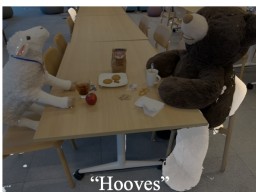

"Chair under the sheep"    "Hooves"

*Figure 8.* Failure cases of LaGa, with prompts shown below and predicted regions highlighted.

clusters $k$, and "Adaptive" represents our final descriptor extraction method, which adaptively selects the number of clusters. "DW" denotes the descriptor weighting scheme.

By replacing the descriptor extraction method with the naive "average pooling" and "max pooling" strategies, the results show significant drops of 10.6% mIoU and 20.7% mIoU, respectively. This is because that merely averaging the multi-view features can lead to information loss due to view-dependent semantics. Conversely, retaining all features makes the segmentation susceptible to noisy features extracted by CLIP. In contrast, using a clustering algorithm to preserve multi-view semantic information leads to a noticeable performance improvement. However, it is evident that the optimal number of clusters varies across different scenes due to differences in data distribution. Manually adjusting these hyper-parameters is impractical and unreasonable, demonstrating the necessity of our adaptive strategy.

Moreover, regardless of whether a fixed number of clusters or the adaptive strategy is used, results with the descriptor weighting consistently outperform those without it. DW$^c$ and DW$^d$ represent descriptor weighting schemes that consider only the internal compactness of the descriptor ($||\mathbf{d}||_2$) and the global alignment ($\langle \mathbf{d}, \bar{\mathbf{v}}^{\mathcal{S}_i} \rangle$), respectively. Both schemes show improvements compared to the –DW setting; however, combining them yields the most significant enhancement, highlighting the necessity of our design.

For more discussion on the hyper-parameter selection of LaGa, please refer to Appendix B.5.

### 6.5. Discussion on Failure Cases

We analyze common failure cases of LaGa. As shown in Figure 8, two main issues are observed:

- **Bag-of-Words Effect in CLIP.** Prompts such as "pirate hat on the rubber duck" or "cookies in the plate" require models to resolve compositional semantics. However, CLIP tends to activate on isolated nouns (e.g., "hat", "duck") rather than the full phrase. LaGa inherits this limitation from CLIP.

- **Lack of Context in 2D Semantics.** LaGa extracts 2D semantics by feeding SAM-segmented image crops into CLIP to reduce distractions from unrelated content. However, this operation removes necessary contextual information, leading to errors in relational ("chair under the sheep") or part-level queries ("hooves").

These failure cases highlight the gap between real-world open-vocabulary perception and current model capabilities. Future work could explore the use of large language models to better handle such compositional grounding.

### 7. Conclusion

In this paper, we investigate language-driven open-vocabulary scene understanding methods based on 3D-GS and identify the view-dependency of 3D semantics as a critical limitation hindering robust 3D understanding. Through experiments, we validate the existence and impact of this issue. To address this challenge, we propose LaGa, a method that establishes cross-view connections among multi-view semantics and constructs view-aggregated semantic representations for 3D scene understanding. Extensive experiments demonstrate that LaGa effectively achieves a more comprehensive understanding of 3D scenes. Our findings provide a new perspective for advancing open-vocabulary scene understanding within the 3D-GS framework.

# Acknowledgments

This work was supported by NSFC 62322604, NSFC 62176159, Shanghai Municipal Science and Technology Major Project 2021SHZDZX0102.

# Impact Statement

This paper presents work whose goal is to advance the field of Machine Learning. There are many potential societal consequences of our work, none which we feel must be specifically highlighted here.

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

## A. Appendix Overview

This appendix provides additional details and analyses to complement the main paper. It includes implementation details, experimental setup, and extensive qualitative results. The appendix is structured as follows:

- Appendix B: Implementation details, including data preparation, additional training strategies, semantic retrieval analysis, cross-view descriptor extraction, hyper-parameter discussion and ScanNet experiments.

- Appendix C.1: Additional qualitative comparisons on the LERF-OVS dataset.

- Appendix C.2: Multi-view segmentation results on the LERF-OVS dataset.

- Appendix C.3: Multi-view segmentation results on the 3D-OVS dataset.

- Appendix C.4: Multi-granularity segmentation results on the LERF-OVS dataset.

- Appendix C.5: Scene editing examples.

- Appendix C.6: Examples of mask groups obtained from 3D scene decomposition.

- Appendix C.7: Analysis of view-dependent 3D semantics, demonstrating variations in 2D relevance scores across different viewpoints.

## B. Implementation Details

To extract 2D segmentation priors and language features for the training images, we follow previous works (Qin et al., 2024; Bhalgat et al., 2024; Wu et al., 2024b) by using the ViT-H model of SAM and the OpenCLIP ViT-B/16 model of CLIP. The semantic feature dimension is $512$, and the Gaussian affinity feature dimension $C'$ is set to 32. For the LERF-OVS dataset, we follow LangSplat to train a three-level model corresponding to the "subpart," "part," and "whole" levels of SAM-extracted masks. Rather than selecting the level with the highest response as in LangSplat, we average responses across all three levels, which we find to be more robust. For 3D-OVS and ScanNet, we use only the "whole" level of SAM masks, as these datasets are simpler. For each scene, the 3D-GS model is trained for 30000 iterations, followed by 30000 iterations training of the Gaussian affinity features. For ScanNet, we apply a KNN-based local feature smoothing operation following SAGA (Cen et al., 2025a) during training the affinity features. During inference, in addition to the relevance score, we find that applying an auxiliary cosine similarity threshold (0.23) helps remove unwanted regions. For all remained objects in the scene, relevance scores are first min-max normalized. A 3D bilateral filtering step is then applied to the resulting 3D relevance map to suppress noise. Gaussians with relevance scores above 0.6 are classified as foreground[3]. All experiments are conducted on a single NVIDIA RTX 3090 GPU.

### B.1. Concrete Data Preparation Pipeline

We follow LangSplat (Qin et al., 2024) to combine SAM with CLIP to generate 2D segmentation priors with pixel-wise semantic features. Specifically, for each 2D image $\mathbf{I}$, we first use SAM to extract a set of 2D masks automatically, denoted by $\mathcal{M}^{\mathbf{I}} = \{\mathbf{M}_i^{\mathbf{I}} \in \{0,1\}^{HW} \mid i = 1, \ldots, N_{\mathbf{I}}\}$. To obtain the corresponding visual semantic feature for a mask $\mathbf{M}_i^{\mathbf{I}}$, we begin by computing its bounding box $\mathbf{B} \in \mathbb{R}^4$. We then apply $\mathbf{M}_i^{\mathbf{I}}$ to $\mathbf{I}$ and crop the resulting masked image with $\mathbf{B}$, producing a cropped image $\mathbf{I}^{\mathbf{M}_i^{\mathbf{I}}}$. Finally, we resize this crop to $224 \times 224$ and feed it into the CLIP image encoder to obtain its visual feature $\mathbf{v}^{\mathbf{M}^{\mathbf{I}}}$.

### B.2. Additional Training Strategy

Due to severe data imbalance when training Gaussian affinity features, we resample positive and negative samples in each training iteration. Rather than applying the loss function in Equation (4) directly, we split it into two components: a positive component that draws features within the same mask closer to the feature prototype $\hat{\mathbf{f}}_{\mathbf{M}}$, and a negative component that pushes features from outside the mask away from the prototype. We randomly select a set of positive samples $\mathcal{P}^{\mathbf{M}}$ and negative samples $\mathcal{N}^{\mathbf{M}}$, ensuring $|\mathcal{P}^{\mathbf{M}}| = |\mathcal{N}^{\mathbf{M}}|$. The rebalanced loss is then defined as:

---

[3]This strategy follows LangSplat, which applies min-max normalization to the 2D relevance map and uses a threshold for foreground-background separation. In contrast, LaGa operates in 3D and applies the same procedure to the 3D relevance map.

$$\mathcal{L}_r = \sum_{\mathbf{I}\in\mathcal{I}} \sum_{\mathbf{M}\in\mathcal{M}^{\mathbf{I}}} \sum_{\mathbf{p}\in\mathcal{P}^{\mathbf{M}}} -\mathbf{M}^{\mathbf{I}}(\mathbf{p})\langle\hat{\mathbf{f}}_{\mathbf{M}}, \mathbf{F}_{\mathbf{I}}(\mathbf{p})\rangle + \sum_{\mathbf{I}\in\mathcal{I}} \sum_{\mathbf{M}\in\mathcal{M}^{\mathbf{I}}} \sum_{\mathbf{p}\in\mathcal{N}^{\mathbf{M}}} \left(1 - \mathbf{M}^{\mathbf{I}}(\mathbf{p})\right) \max(\langle\hat{\mathbf{f}}_{\mathbf{M}}, \mathbf{F}_{\mathbf{I}}(\mathbf{p})\rangle, 0). \tag{13}$$

This re-balancing strategy is essential for obtaining high-quality scene decomposition results.

In addition, to better align affinity features between inner Gaussians and those located near the object surface, we follow SAGA (Cen et al., 2025a) and incorporate a feature norm regularization. Specifically, the aggregated feature $\mathbf{F}(\mathbf{p})$ at location $\mathbf{p}$ is computed as:

$$\mathbf{F}(\mathbf{p}) = \sum_{i=1}^{|\mathcal{G}_{\mathbf{p}}|} \frac{\mathbf{f}_{\mathbf{g}_i^{\mathbf{p}}}}{||\mathbf{f}_{\mathbf{g}_i^{\mathbf{p}}}||_2} \alpha_{\mathbf{g}_i^{\mathbf{p}}} \prod_{j=1}^{i-1} (1 - \alpha_{\mathbf{g}_j^{\mathbf{p}}}), \tag{14}$$

and the regularization term is defined as:

$$\mathcal{L}_{\text{norm}}(\mathbf{p}) = 1 - ||\mathbf{F}(\mathbf{p})||_2, \tag{15}$$

resulting in the final loss function:

$$\mathcal{L} = \mathcal{L}_r + \sum_{\mathbf{I}\in\mathcal{I}} \sum_{\mathbf{p}\in\delta(\mathbf{I})} \mathcal{L}_{\text{norm}}(\mathbf{p}). \tag{16}$$

Please refer to Cen et al. (2025a) for more details on the underlying mechanism of this regularization.

### B.3. Details of Semantic Retrieval

To obtain the fused semantic features of 3D Gaussians, we first analyze the 3D Gaussians responsible for rendering each 2D mask. This is achieved by examining the weights used in the rendering phase (Equation (1)). For ease of implementation, given a 2D mask $\mathbf{M}$, we use the backward pass of the differentiable rasterization algorithm to propagate the mask information onto the 3D Gaussians. This process yields a set of gradient scores $\mathcal{Z} = \{z_{\mathbf{g}} \mid \mathbf{g}\in\mathcal{G}\}$. Let $\mathbf{l}_{\mathbf{g}}$ denote the fused visual feature from the semantic retrieval experiment. During an iteration of projection, $\mathbf{l}_{\mathbf{g}}$ is updated as follows:

$$\mathbf{l}_{\mathbf{g}} \leftarrow \mathbf{l}_{\mathbf{g}} - z_{\mathbf{g}}\mathbf{v}^{\mathbf{M}}, \tag{17}$$

where $\mathbf{v}^{\mathbf{M}}$ represents the feature of the 2D mask $\mathbf{M}$. After processing all mask features, the fused visual features are normalized:

$$\mathbf{l}_{\mathbf{g}} \leftarrow \frac{\mathbf{l}_{\mathbf{g}}}{||\mathbf{l}_{\mathbf{g}}||_2}. \tag{18}$$

Using these fused features, the 3D Gaussians corresponding to a given 2D mask feature $\mathbf{v}^{\mathbf{M}}$ are retrieved as:

$$\{\mathbf{g} \mid \mathbf{g}\in\mathcal{G}, \langle\mathbf{v}^{\mathbf{M}}, \mathbf{l}_{\mathbf{g}}\rangle > 0.75\}. \tag{19}$$

Note that the 0.75 threshold is chosen empirically based on analysis of the precision–recall trade-off under different cosine similarity values. Here, precision refers to the proportion of retrieved 3D Gaussians that actually belong to the corresponding 3D object. Low precision indicates that unrelated Gaussians are being retrieved for a given 2D mask.

As shown in Table 5, lowering the similarity threshold (e.g., to 0.7) increases the proportion of samples with high recall (¿0.9), but at the cost of significantly reduced precision. For example, at 0.7, the average precision of high-recall samples drops to just 13.4%.

From these observations, we find that thresholds in the [0.75, 0.8] range strike a better balance. We conservatively select 0.75, where 50.2% of samples exhibit low recall, clearly demonstrating the challenge of view-dependent semantics. A threshold of 0.8 also yields valid results, with fewer high-recall samples but higher precision.

Note that the average precision does not reach 100%, as CLIP features operate at the semantic level, not the instance level. Therefore, 3D Gaussians with similar semantics but belonging to different objects may also be retrieved, even with a high threshold.

*Table 5.* Precision-recall trade-off at different relevance thresholds.

| Threshold | 0.5 | 0.6 | 0.7 | 0.75 | 0.8 | 0.9 |
|---|---|---|---|---|---|---|
| Low / (Low+High) (%) | 2.6 | 13.1 | 33.7 | 50.2 | 65.4 | 93.8 |
| AP of High (%) | 1.7 | 3.9 | 13.4 | 24.1 | 38.6 | 48.7 |

---

**Algorithm 1** Cross-View Descriptor Extraction

---

**Input:** Feature set $\mathcal{V}^{\mathcal{S}_i}$, maximum clusters $K_{\max}$
**Output:** Semantic descriptors $\mathcal{D}^{\mathcal{G}^{\mathcal{S}_i}}$ for the 3D object $\mathcal{G}^{\mathcal{S}_i}$, and the number of descriptors $N^{\mathcal{G}^{\mathcal{S}_i}}$.
Initialize best silhouette score $s^* \leftarrow -1$
Initialize optimal number of clusters $K^* \leftarrow 1$
**for** $K = 1$ **to** $K_{\max}$ **do**
 Perform K-means clustering on $\mathcal{V}^{\mathcal{S}_i}$ with $K$ clusters, obtaining clusters $\mathcal{C}_1, \ldots, \mathcal{C}_K$
 Compute silhouette score $s_K$ for the current clustering
 **if** $s_K > s^*$ **then**
  $s^* \leftarrow s_K,\ K^* \leftarrow K$
  $\mathcal{C}^* \leftarrow \{\mathcal{C}_1, \ldots, \mathcal{C}_K\}$
 **end if**
**end for**
$N^{\mathcal{G}^{\mathcal{S}_i}} \leftarrow K^*$
$\mathcal{D}^{\mathcal{G}^{\mathcal{S}_i}} \leftarrow \{\texttt{Centroid}(\mathcal{C}_1^*), \ldots, \texttt{Centroid}(\mathcal{C}_{K^*}^*)\}$

---

### B.4. Detailed Algorithm of the Cross-View Descriptor Extraction

The pseudo code of the cross-view descriptor extraction is shown in Algorithm 1.

### B.5. Hyper-parameter Selection and Discussion

LaGa involves two key hyper-parameters: the maximum number of clusters ($K_{\max}$) in the adaptive K-means algorithm, and the cluster selection threshold $\epsilon$ used in HDBSCAN. $K_{\max}$ is set to 20 for all experiments. For multi-level modeling (see Appendix B.1), $\epsilon$ is set to 0.1, 0.2, and 0.3 for the "subpart", "part", and "whole" levels, respectively. In experiments with a single-level decomposition, $\epsilon$ is fixed at 0.1.

To evaluate the robustness of LaGa with respect to these hyper-parameters, we conduct ablation studies on the LERF-OVS dataset. As shown in Table 6, LaGa maintains stable performance across a wide range of $K_{\max}$ values (5–30). Similarly, Table 7 demonstrates that LaGa is insensitive to $\epsilon$ within the range of 0 to 0.3 at the "whole" level. However, setting $\epsilon$ too large (*e.g.*, 0.4) can lead to unintended object merging.

### B.6. Implementation Details for Experiments on ScanNet Dataset

For the point cloud semantic segmentation task on the ScanNet dataset, we adapt our 3D segmentation method to align with the evaluation protocol, which does not support a complete 3D-GS model. Specifically, during the cross-view semantics grouping phase, we fix the coordinates of the 3D points (i.e., the means of the 3D Gaussians) and disable the densification mechanism of 3D-GS. This allows us to directly train the affinity features of these Gaussians while maintaining consistency in other processes. During evaluation, we employ the inference procedure described in Section 5.4 to assign each 3D point a multi-class segmentation score, classifying each point into the category with the highest score.

## C. More Qualitative Results

In this section, we present additional visual results to further demonstrate the effectiveness of LaGa. We first show more qualitative comparisons with existing methods on the LERF-OVS dataset. Next, we provide multi-view 3D segmentation results on LERF-OVS to demonstrate the integrity of the segmented objects of LaGa. We also show multi-view semantic

Table 6. Impact of $K_{\max}$ on segmentation performance (mIoU).

| $K_{\max}$ | 5 | 10 | 15 | 20 | 30 |
|---|---|---|---|---|---|
| mIoU (%) | 63.4 | 63.4 | 64.1 | 64.0 | 63.2 |

Table 7. Effect of $\epsilon$ on segmentation performance (mIoU).

| $\epsilon$ | 0 | 0.1 | 0.2 | 0.3 | 0.4 |
|---|---|---|---|---|---|
| mIoU (%) | 62.6 | 63.0 | 62.1 | 64.0 | 60.6 |

segmentation results on 3D-OVS. Finally, we offer some representative scene-editing examples to highlight the practical significance of LaGa.

### C.1. More Visual Comparisons with Existing Methods

Figure 9 presents the results, which lead to conclusions similar to those in Section 6.3. One phenomenon worth clarifying is the needle-like borders around segmented objects. This effect arises from an inherent flaw of the 3D-GS representation, where 3D Gaussians are trained to fit multi-view RGB images without explicit object awareness. Consequently, numerous ambiguous Gaussians contribute to the rendering of different objects. After segmentation, these Gaussians manifest as needle-like borders. Although certain 3D-GS segmentation methods (Cen et al., 2025b; Hu et al., 2024) have examined this issue, it lies beyond the scope of this paper.

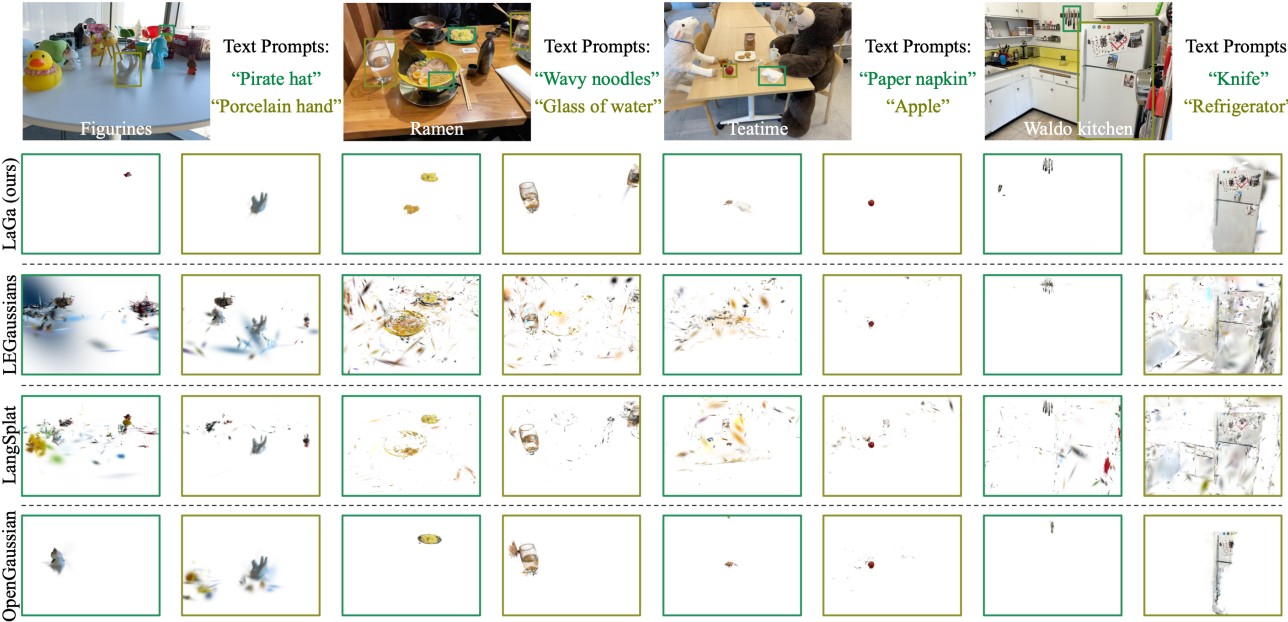

Figure 9. More qualitative comparison results on the LERF-OVS dataset.

### C.2. Multi-view Segmentation Results on the LERF-OVS Dataset

We present multi-view segmentation results for the LERF-OVS dataset in Figure 10. Both rendered 2D masks and the corresponding 3D objects (with backgrounds removed) are displayed. In the "Figurines–pink ice cream" and "Teatime–coffee mug" examples, the 2D rendered masks are empty, yet the 3D objects become visible once occlusions are removed. This underscores a key difference between previous 2D-based approaches and our 3D method: because 2D methods rely on rendered feature maps, they cannot perceive objects occluded by obstacles. In contrast, conducting direct 3D understanding in 3D space is free from this limitation.

### C.3. Multi-view Semantic Segmentation Results on the 3D-OVS Dataset

We also present multi-view semantic segmentation results on the 3D-OVS dataset, as shown in Figure 11. Although LaGa is primarily designed for 3D space, it readily handles forward-facing scenes. The forward-facing nature of this dataset reduces the impact of view-dependent semantics. Nevertheless, LaGa achieves on-par performance with existing 2D-based methods.

### C.4. Multi-granularity Segmentation Results on the LERF-OVS Dataset

As described in Appendix B, LaGa performs multi-granularity segmentation by training three levels of affinity features. We show this ability in Figure 12. The "Avg." setting denotes the default strategy that averages predictions across all levels. We observe that relying on a single segmentation level is generally less reliable than the multi-level averaging approach. At the coarsest level, different objects may occasionally be merged due to under-segmentation by SAM. At the finest ("subpart") level, over-segmentation often produces regions with limited semantic content, making CLIP-based classification less effective.

### C.5. Scene Editing Results

We present a few representative scene-editing examples in Figure 13. By selecting 3D objects via text prompts, we can modify their appearance or remove them entirely from the scene. A more intriguing example involves transplanting a "cup" from the "Figurines" scene into the "Waldo-kitchen" scene. A more intriguing example involves transplanting a "cup" from the "Figurines" scene into the "Waldo-kitchen" scene. This demonstration not only confirms LaGa's ability to retrieve high-quality objects but also highlights its potential for more complex interactive tasks, such as training AI agents by constructing simulation environments.

### C.6. Mask Group Examples

We present visualization examples of mask groups obtained during the cross-view semantics grouping phase, as shown in Figure 14. Masks representing the same 3D object are accurately grouped regardless of their relative size in the current view or the viewing angle, enabling a comprehensive aggregation of view-dependent semantics.

In addition to demonstrating the effectiveness of the semantics grouping approach, we aim to uncover the general causes of view-dependent semantics by analyzing the multi-view masked images. We attribute view-dependent semantics primarily to key information loss under specific viewpoints. For instance, targets of interest may become occluded, fall outside the field of view, or appear too distant for their key features to be recognized. These scenarios are both common and unavoidable due to the inherent limitations of the vision cone. Another contributing factor is poor data quality, as illustrated by the spice jars in the 'Waldo_Kitchen' scene. Blurred images in the collected dataset result in a loss of detail for concrete objects. While the former issue is intrinsic, the latter can be mitigated through more careful data collection processes.

### C.7. Examples of View-Dependent Semantics

To further illustrate the phenomenon of view-dependent semantics, we present qualitative results in Figure 15. The relevance score for each 2D semantic feature is computed using Equation (10), where scores above 0.5 are classified as positive. The computed relevance scores are displayed above each figure, with the corresponding text prompts shown in the top-left corner of each row.

Notably, the semantics of the same 3D object exhibit significant variations across different viewpoints. Even when the 2D masks are correctly assigned, the relevance scores for the same 3D object can range from 0.3 to 0.8. This variation strongly demonstrates the existence of significant view-dependent semantics.

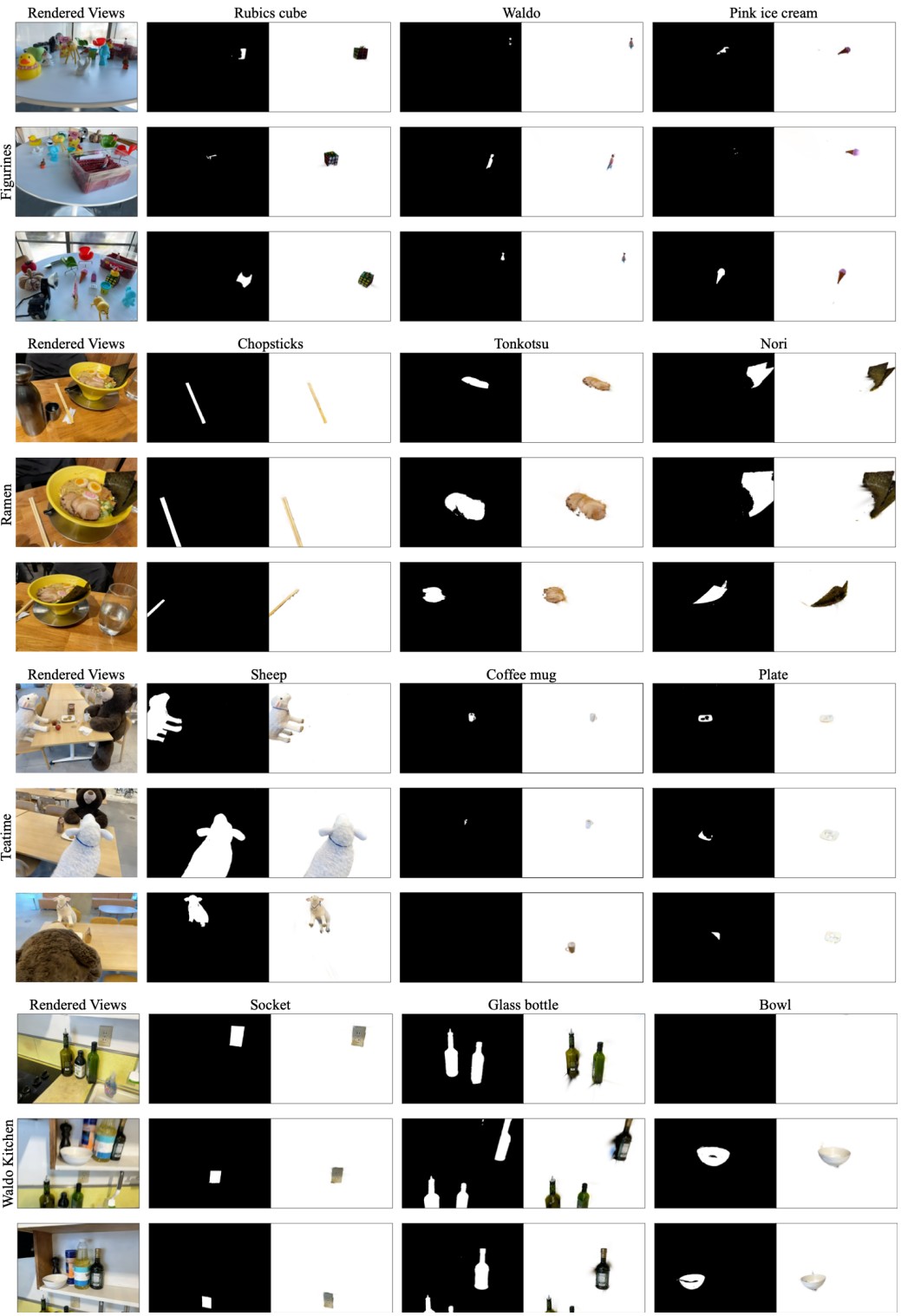

*Figure 10.* Multi-view segmentation results on the LERF-OVS dataset.

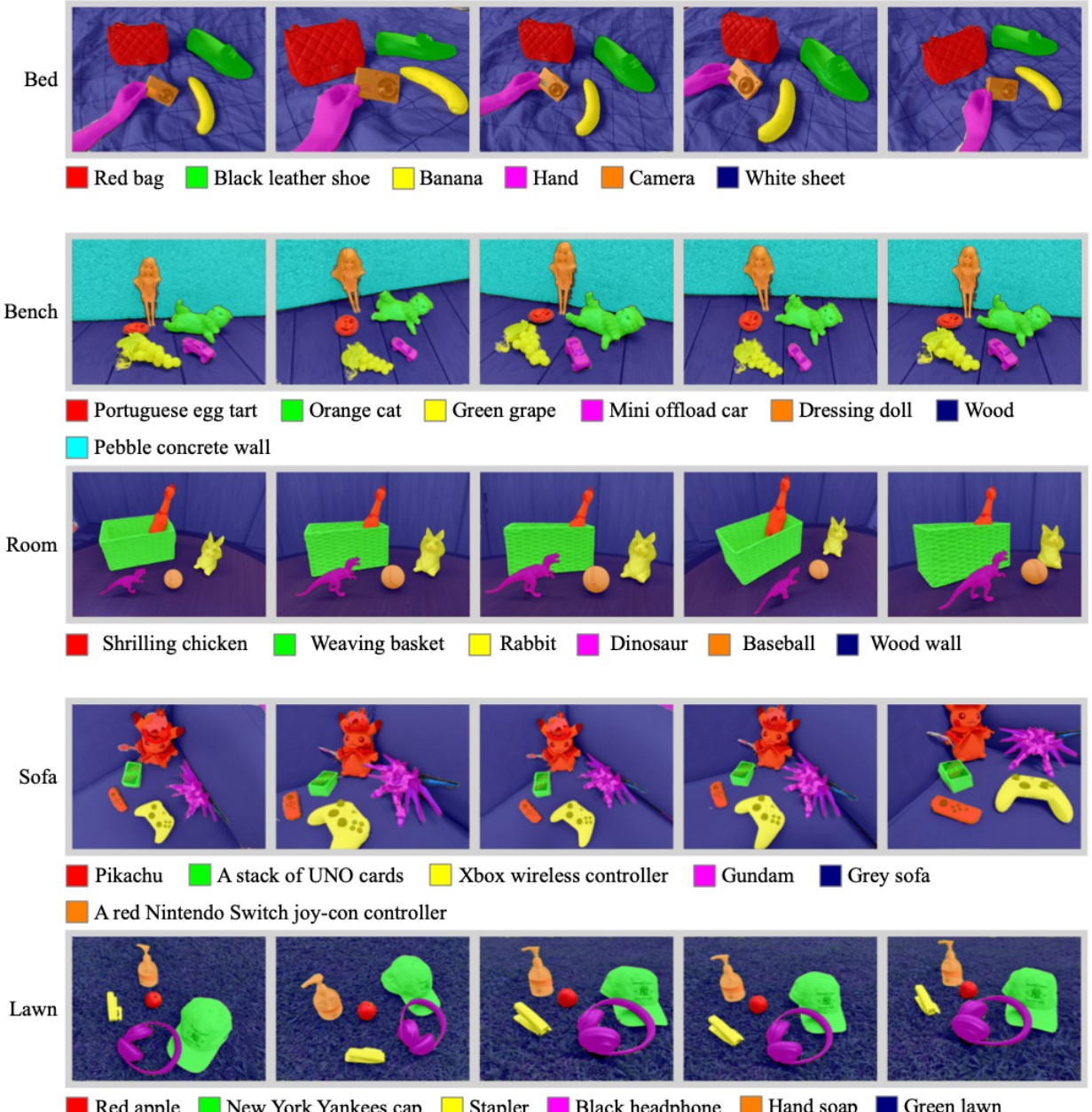

*Figure 11.* Multi-view semantic segmentation results on the 3D-OVS dataset.

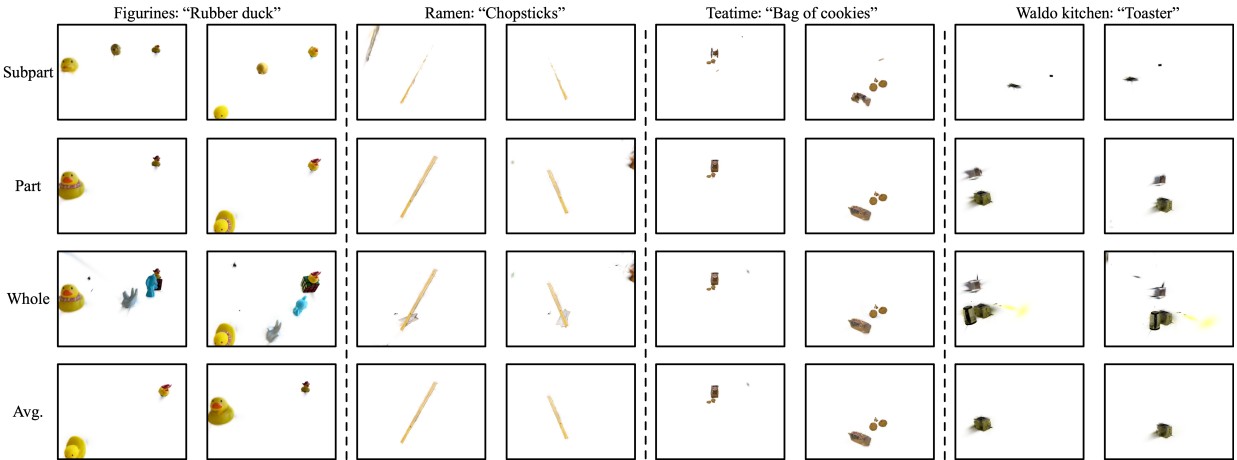

*Figure 12.* Multi-granularity semantic segmentation results on the LERF-OVS dataset.

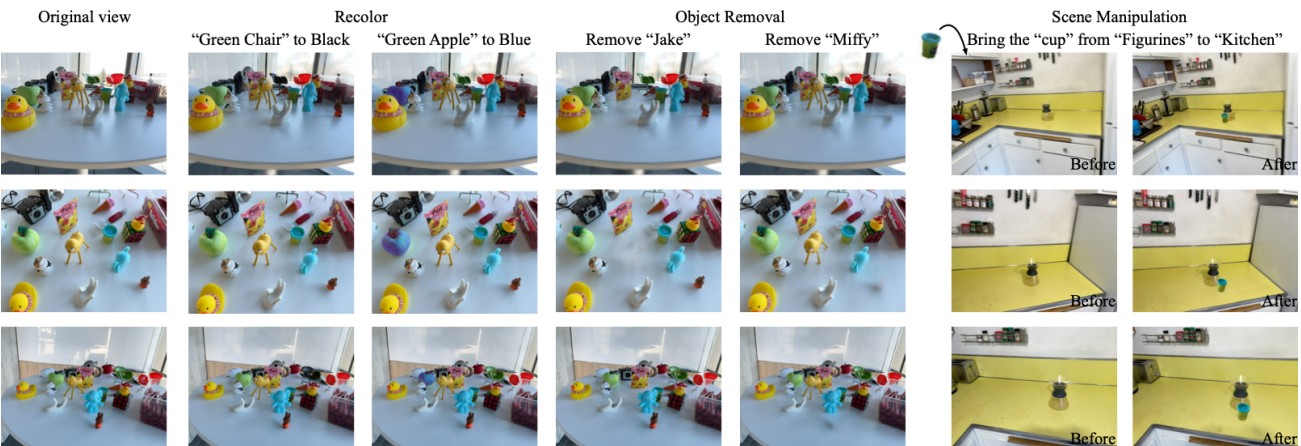

*Figure 13.* Scene editing examples with the help of LaGa.

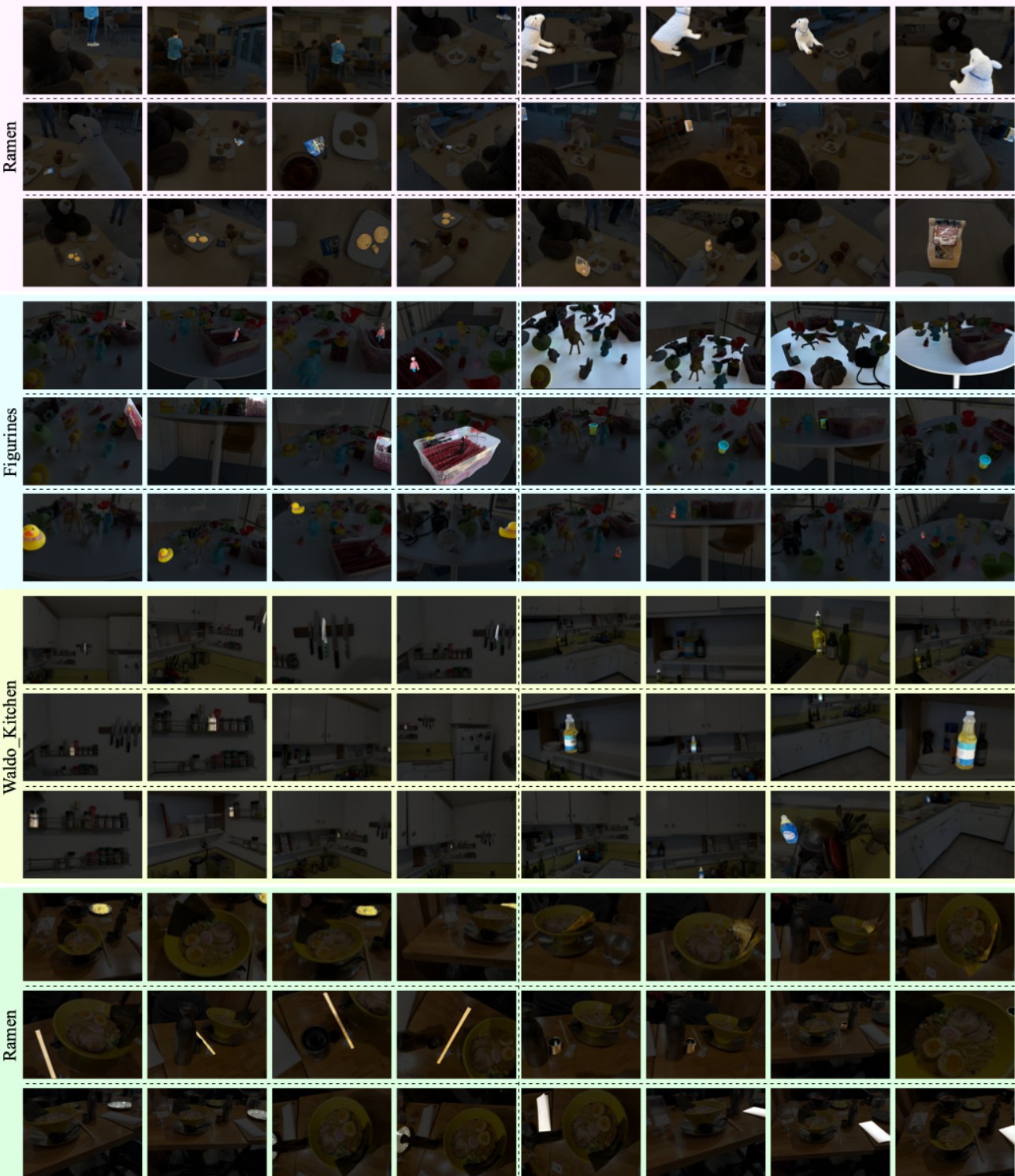

*Figure 14.* Visualization of mask groups obtained from the 3D scene decomposition phase. Highlighted regions indicate the corresponding masks. Groups of masks are visually separated by dotted lines.

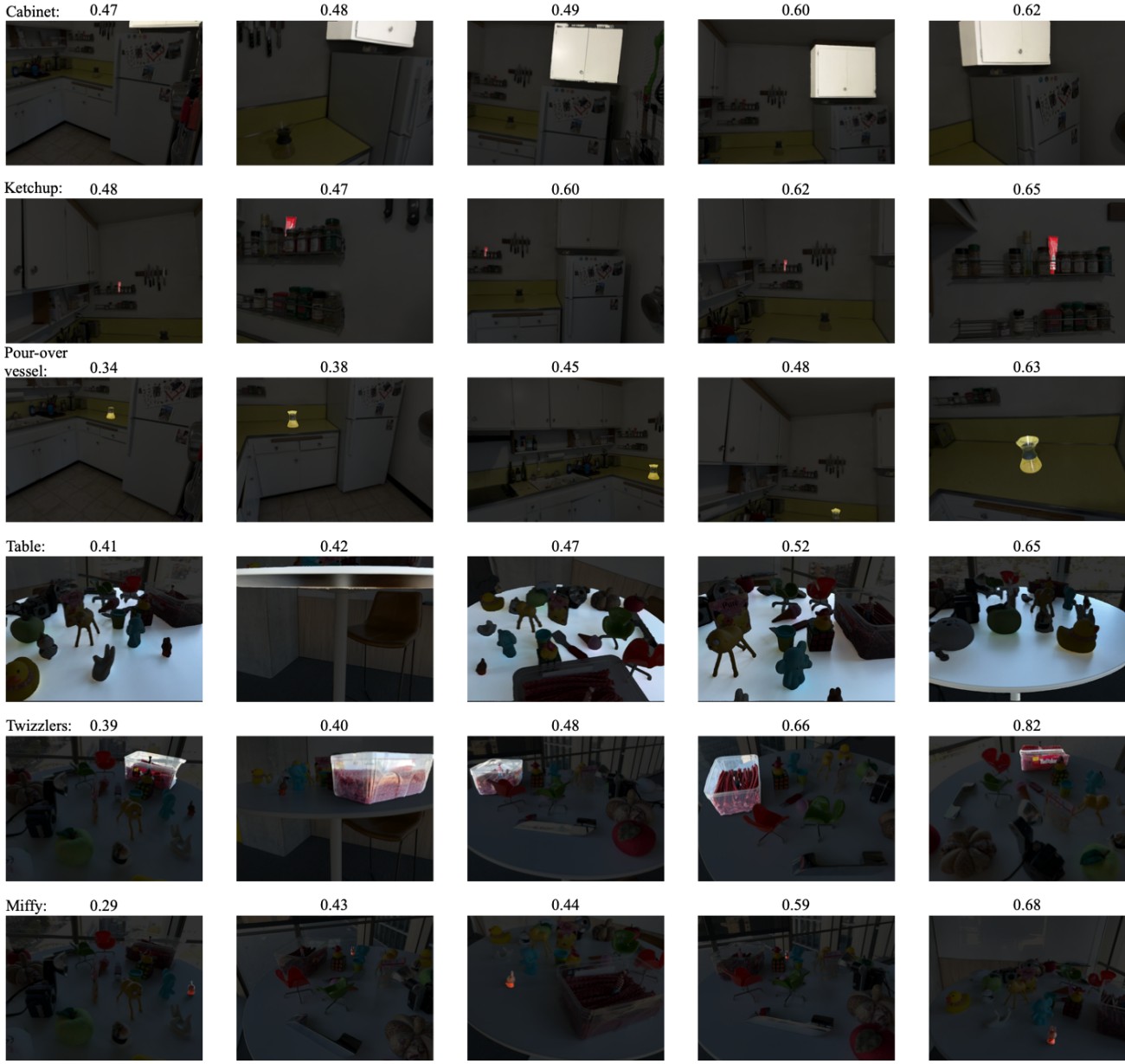

*Figure 15.* View-dependent semantic variations within the same mask group. Each row corresponds to a different text prompt, shown in the top-left corner. The relevance scores for individual 2D semantics are displayed above each figure.

