# OpenReview forum: "Tackling View-Dependent Semantics in 3D Language Gaussian Splatting"
_ICML.cc/2025/Conference — ICML 2025 poster_

### Official Review · Reviewer_YHPs · 2025-02-17

**Overall Recommendation:** 4

**Summary:**

This paper proposes a novel method，named LaGA and it tackles the challenge of view-dependent semantics in language-driven open-vocabulary 3D scene understanding.
 LaGa decomposes the 3D scene into distinct objects and then builds view-aggregated semantic representations by clustering semantic descriptors and reweighting them based on multi-view information. Extensive experiments demonstrate that this approach effectively captures the key nuances of view-dependent semantics, leading to a more comprehensive understanding of 3D scenes.

**Claims And Evidence:**

yes

**Essential References Not Discussed:**

No.

**Experimental Designs Or Analyses:**

Extensive experiments demonstrate that LaGa effectively captures key information from view-dependent semantics, enabling a more comprehensive understanding of 3D scenes.

**Methods And Evaluation Criteria:**

yes, it does make sense.

**Other Comments Or Suggestions:**

In addition to the quantitative comparisons for 3D open-vocabulary localization, I would also like to see 2D query results, as well as qualitative results on the mipnerf360 dataset— for example, for “Room” and “Garden”.

**Other Strengths And Weaknesses:**

### Strengths
This paper is written clearly, with well-articulated motivation and clearly presented figures and charts. The experiments demonstrate that the proposed method can achieve precise open-vocabulary 3D Gaussian localization.
### Weaknesses
The claimed innovation in 3D Scene Decomposition shows notable similarities with the previous work on openGaussian[1]. I recommend that the authors explicitly discuss these similarities and clarify how their approach differs from and improves upon the prior work.


[1] OpenGaussian: Towards Point-Level 3D Gaussian-based Open Vocabulary Understanding

**Questions For Authors:**

No

**Relation To Broader Scientific Literature:**

It is worth noting that the claimed innovation in 3D Scene Decomposition shows notable similarities with the previous work on openGaussian[1]. I recommend that the authors explicitly discuss these similarities and clarify how their approach differs from and improves upon the prior work.

[1] OpenGaussian: Towards Point-Level 3D Gaussian-based Open Vocabulary Understanding

**Theoretical Claims:**

Yes, I have reviewed the theoretical explanations provided in the paper for the following components: 3D Scene Decomposition, View-Aggregated Semantic Representation, and Weighted Descriptor Relevance Aggregation. I did not find any errors in these sections.

---

> ### Author Rebuttal · Authors · 2025-04-01
>
> We sincerely thank you for your constructive comments. We hope our response can help address your concern.
>
> ## Weaknesses
>
> >W1: The claimed innovation in 3D Scene Decomposition shows notable similarities with the previous work on openGaussian. I recommend that the authors explicitly discuss these similarities and clarify how their approach differs from and improves upon the prior work.
>
> Thanks for the insightful suggestion. We discuss the relationship between OpenGaussian and LaGa below:
>
> **Similarity:**
> Both OpenGaussian and LaGa first decompose a 3D scene into objects before associating the scene with semantic information.
>
> **Key Differences in Motivation:**
> The two methods are built upon fundamentally different motivations. OpenGaussian addresses the inconsistency between 3D point-level features and 2D pixel-level CLIP features, attributing it to
> (1) inaccurate 2D–3D associations introduced by alpha-blending during differentiable rendering, and
> (2) limited expressiveness from 3D feature compression for efficinet rendering.
> In contrast, LaGa focuses on tackling the semantic discrepancy among multi-view observations of the same 3D object, which naturally arises in open-vocabulary 3D understanding. This shift in perspective leads to substantially different methodologies.
>
> **Key Differences in Methodology:**
> To mitigate 2D–3D feature inconsistency, OpenGaussian decomposes the 3D scene so that it can directly assign a 2D CLIP feature to each Gaussian of an object. It employs a rule-based representative view selection strategy for this assignment. While straightforward, this strategy neglects rich information embedded in multi-view semantics and lead to suboptimal performance.
>
> In contrast, LaGa extracts semantic descriptors across views using adaptive clustering and then adopts the weighted relevance aggregation to suppress noisy semantics and enhance robustness. This design enables LaGa to construct a more comprehensive and robust 3D semantic representation. Notably, LaGa does not involve any 3D vision-language feature training, and thus is also unaffected by the 2D–3D inconsistency problem that OpenGaussian aims to address. The design of LaGa enables it to tackle complex 3D objects with various multi-view semantics, evidenced by its significant +18.7% mIoU improvement over OpenGaussian.
>
> **On Scene Decomposition:**
> We acknowledge that our scene decomposition strategy is not the core innovation of LaGa, which is inspired by prior contrastive-learning-based approaches. However, compared to OpenGaussian’s two-stage codebook-based decompsition pipeline, which requires manual tuning of two codebook sizes and thus incurs higher computational cost, LaGa adopts a more lightweight solution, i.e., using HDBSCAN to automatically determine the number of decomposed objects based on feature density. This results in comparable decomposition accuracy while significantly reducing system complexity.
>
> We will incorporate this clarification and discussion in the revised paper for improved clarity and completeness.
>
>
> >W2: In addition to the quantitative comparisons for 3D open-vocabulary localization, I would also like to see 2D query results, as well as qualitative results on the mipnerf360 dataset— for example, for “Room” and “Garden”.
>
> Thank you for the valuable suggestion. We provide several 2D segmentation masks in Figure 8, which demonstrate the 2D projections of LaGa’s 3D query results. Since LaGa does not involve any 2D CLIP feature learning, it is not intended for direct querying on the 2D image plane. Nevertheless, we believe that our 3D-centric paradigm can serve as a more flexible alternative to 2D methods, as the 3D query result can be rendered from arbitrary viewpoints without requiring repeated queries for each individual image.
>
> In addition, we have conducted qualitative experiments on the MIP-360 dataset, including scenes such as Room and Garden, and observed that LaGa generalizes well to these scenes. The corresponding visualizations will be included in the revised paper.

---

### Official Review · Reviewer_sKQC · 2025-03-02

**Overall Recommendation:** 3

**Summary:**

The paper addresses the challenge of view-dependent semantics in 3D Gaussian Splatting for language-driven open-vocabulary scene understanding. The authors propose LaGa (Language Gaussians), a method that decomposes 3D scenes into objects and constructs view-aggregated semantic representations by clustering and reweighting semantic descriptors based on multi-view semantics. Specifically, they weigh the descriptor relevance with the directional consistency and internal compactness. The paper claims significant improvements over state-of-the-art methods, particularly on the LERF-OVS dataset, with a +18.7% mIoU improvement.

##Update after rebuttal
I thank the authors for their clarification. I have no further concerns, but I remain neutral regarding this paper's novelty and significance. Therefore, I have decided to maintain my score as weak accept.

**Claims And Evidence:**

The claims made in the paper are generally supported by clear and convincing evidence. The authors provide extensive experiments to validate their motivation, including semantic similarity distribution analysis and semantic retrieval integrity analysis. The quantitative results on the LERF-OVS, 3D-OVS, and ScanNet datasets show improvements over existing methods, supporting the claim that LaGa achieves more comprehensive 3D scene understanding.

**Essential References Not Discussed:**

The references seem sufficient.

**Experimental Designs Or Analyses:**

The experimental designs and analyses are sound and well-executed. The authors conduct a thorough evaluation of their method on multiple datasets and provide both quantitative and qualitative results.

**Methods And Evaluation Criteria:**

The proposed methods make sense for the problem at hand. LaGa's approach to decomposing 3D scenes into objects and constructing view-aggregated semantic representations is a logical way to address view-dependent semantics. The evaluation criteria are appropriate, with the authors using standard benchmarks (LERF-OVS, 3D-OVS, and ScanNet) to evaluate their method. The use of mIoU as a metric is standard in semantic segmentation tasks and provides a clear measure of performance.

**Other Comments Or Suggestions:**

No other comments.

**Other Strengths And Weaknesses:**

Strengths:
1. The paper is well-motivated, aiming to address the inconsistency of the 2D CLIP embedding.
2. The experimental results are strong, with improvements over state-of-the-art methods on multiple datasets.

Weaknesses:
1. The proposed method does not support multi-scale segmentation like LangSplat, which could hinder its practicality in open-vocabulary segmentation.
2. Although the proposed method shows strong results, it is more engineering-driven and does not provide enough knowledge advancement. The idea of addressing the feature inconsistency has been proposed in previous work such as OpenGaussian.

**Questions For Authors:**

1. Can the method perform multi-scale segmentation?
2. What are the runtime statistics of this method (e.g. how long and how much memory does it take)?

**Relation To Broader Scientific Literature:**

This paper aims to resolve the inconsistency of the 2D language embeddings of CLIP, which is an important task as currently we only have 2D multimodal foundation models and can only obtain 3D features by distilling features from 2D models. Thus the 3D inconsistency of 2D models is an important problem to tackle.

**Theoretical Claims:**

The paper does not present any theoretical proof, so there are no theoretical claims to evaluate.

---

> ### Author Rebuttal · Authors · 2025-04-01
>
> We thank you for your effort in reviewing our paper. We hope the following response would address your concerns.
>
> ## Weaknesses:
> >W1: ... does not support multi-scale segmentation like LangSplat ...
>
> Thank you for the comment. LaGa supports multi-scale segmentation, though not sufficiently emphasized in the main paper. Some details are provided in Appendix B under the implementation section. Specifically, during the scene decomposition, LaGa maintains three affinity features for each 3D Gaussian, which are supervised using SAM-extracted segmentation masks at the subpart, part, and whole levels, respectively. During inference, the final result is produced by averaging the responses from these three levels, enabling LaGa to capture semantic concepts across varying granularities.
>
> This multi-scale capability is further reflected in the visualization results. For example, in Ramen scene shown in Fig. 6, OpenGaussian merges the 'kamaboko' with the surrounding noodles due to its lack of multi-scale segmentation ability, whereas LaGa successfully distinguishes it. Similarly, in Fig. 7, LaGa accurately segments the pirate hat from the rubber duck. Moreover, we verify that when queried with 'rubber duck', LaGa retrieves both the duck and its pirate hat, demonstrating its effectiveness in capturing hierarchical part-whole semantics.
>
> We will incorporate the visual results of 'rubber duck' into the revised paper, along with more specifically designed visualizations to better illustrate the multi-scale segmentation ability of LaGa.
>
> >W2: ... engineering-driven and does not provide enough knowledge advancement. The idea of addressing the feature inconsistency has been proposed in OpenGaussian.
>
> Thank you for the thoughtful comment. While our method includes practical design choices, we believe it also contributes meaningful insights to open-vocabulary 3D scene understanding. In particular:
>
> 1. Identification and analysis of view-dependent semantics:
> We are the first to systematically identify and quantitatively analyze the phenomenon of view-dependent semantics, as illustrated in Figures 3, 4, and 12. This issue, long overlooked by prior work including OpenGaussian, plays a critical role in open-vocabulary 3D understanding and poses unique challenges for multi-view semantic aggregation.
> 2. An effective approach for semantic aggregation across views:
> We propose an effective approach, LaGa, which is able to robustly suppress noisy or inconsistent semantics while preserving informative signals, without requiring manual rules or task-specific heuristics. We believe the simplicity, generalizability, and effectiveness of our approach can serve as a good foundation for future research in this area.
>
> **Comparison with OpenGaussian:**
>
> While both LaGa and OpenGaussian aim to address feature inconsistency, their motivations and methodologies differ fundamentally.
>
> OpenGaussian focuses on the **inconsistency between 3D point-level features and 2D pixel-level CLIP features**, attributing the problem to (1) inaccurate 2D–3D feature association caused by the alpha-blending in differentiable rendering, and (2) limited expressiveness due to 3D feature compression for rendering efficiency. To address this, OpenGaussian decomposes the 3D scene, so that it can adopt a hand-crafted strategy to directly assign a 2D CLIP feature from a specific view to all 3D Gaussians within an object. However, this hard assignment discards rich multi-view semantic cues, resulting in substantial information loss and ultimately sub-optimal performance.
>
> In contrast, LaGa recognizes feature inconsistency as the **semantic discrepancy among multi-view observations of the same 3D object**, which arises naturally in 3D scene understanding. To address this, LaGa proposes to aggregate view-specific 2D semantics into comprehensive and robust 3D semantic representations through adaptive semantic descriptor extraction and descriptor reweighting. Note that LaGa does not involve any 3D vision-language feature training, and thus is also unaffected by the 2D–3D inconsistency problem that OpenGaussian aims to address.
>
> The revelation of the critical view-dependent semantics issue and the corresponding improvements help LaGa achieve a significant performance gain: LaGa outperforms OpenGaussian by +18.7% mIoU under the same experimental setting, validating the effectiveness of our approach.
>
> ## Questions:
> > Q1: Multi-scale segmentation?
>
> Yes. Please refer to our response to W1.
> > Q2: Runtime statistics of this method (e.g. how long and how much memory does it take)?
>
> LaGa is highly efficient. We evaluate its runtime performance on the LERF-OVS dataset using a single NVIDIA RTX 3090 GPU. The inference time per query ranges from approximately 80 ms to 200 ms, with peak GPU memory usage between 5 GiB and 13 GiB. Note that compared with 2D methods like LangSplat, which takes about 250 ms per query on the rendered feature map, LaGa directly deliver 3D point-wise understanding.

---

### Official Review · Reviewer_QD2y · 2025-03-13

**Overall Recommendation:** 1

**Summary:**

The paper proposes to perform open-vocabulary semantic segmentation of 3DGS scenes that respect view-dependent and view-independent semantics. Using SAM masks, per-gaussian contrastive features are learned to learn 3D object clusters. The method is evaluated on LERF OVS and ScanNet datasets showcasing competitive quality.

**Claims And Evidence:**

The paper claims that prior methods for 3D semantic understanding suffer from view-dependency of 3D semantics which limits their quality  and robustness. The claim is supported by experiments on precision and recall rates for the 2D-3D semantic features.

**Essential References Not Discussed:**

- Table 1 SAGA, GARField are strong baselines to compare against which are missing in the evaluations.

**Experimental Designs Or Analyses:**

N/A

**Methods And Evaluation Criteria:**

The problem is well-motivated with experimental analyses. The core part of the method is learning contrastive features which is similar to SAGA, GARField among many others. The biggest drawback of this method are the steps that builds on top of it. Overall, the clustering and filtering approaches performed seem crude and not that impactful based on quantitative evaluations.

They also seem to require careful tuning of hyperparameters for it to work well. It would be great if the authors can comment on this.

The HDBSCAN performed on top of \hat{f}_M seems to dictate the number of segmented objects in the scene and affect semantic view-aggregation downstream. But how is the number of clusters N_p obtained in the first place? A wrong value could under or over-segment the scene leading to misaligned semantics.

Sensitivity to SAM masks: How do you handle SAM masks that aren't consistent in mask boundaries for a particular object across multiple frames. This is very common artifact in the LERF dataset for \textit{figurines}, \textit{ramen}, and \textit{teatime}.

What is the reason for performing k-means clustering in Cross-View Descriptor Extraction? An alternative is to perform farthest point sampling that will represent the most "diverse" viewpoints. With kmeans there is a possibility of averaging out far away features which might be undesirable.

The quantitative evaluations are performed by rendering the 2D binary map. Its not clear if its completely sufficient to show 3D segmentation quality unless 1) evaluation is performed directly in 3D or 2) 360 degree 2D evaluation is performed. Only table 3 and figure 6 actually support the claim of 3D segmentation quality.

Table 4:
1) why is -DW better than +DW for fixed k=5,10,20 on figurines and teatime?
2) For Waldo kitchen scene with adaptive clustering, -DW performs better than +DW^c and +DW^d. But the combined +DW is better than -DW. How is this possible?
3) For Teatime scene, average pooling is better than fixed k=5,20 but worse than k=10. What is the reasoning for this? Is Avg pooling for k > 20?
4) The adaptive scheme is not a clear winner in any one scene.

**Other Comments Or Suggestions:**

- Eq. 3 should be performed per mask. M^I_{i} and not M^{I}
- Eq. 8 should be C not C' for dimension of d_{i}.
- Eq. 10  exp <d \dot \phi^{i}_{canon}>. \dot -> ,

**Other Strengths And Weaknesses:**

The method follows a well-established approach to segment 3D Gaussians using contrastive losses. By performing clustering and further view-weighted associations on top of the contrastive features, they are robust to viewpoints and retain semantics for all gaussians belonging to a semantic entity. However this would be sensitive to hyperparameters and would require extensive tuning per scene/dataset.

**Questions For Authors:**

- For evaluation, how is the binarized gaussians associated with the 2D ground-truth masks? based on mask iou?
- What is the effect of Cross-View Descriptor Extraction individually as an ablation?
- For internal compactness, couldn't larger clusters have the L2-norm be much greater than 1?
- L632-633: "... follow LangSplat to train a three-layer model...". Do you train a complete Langsplat model (learning clip features on gaussians)? why is it that necessary? It seems quite overkill for the purpose of getting multi-view consistent 2D segmentations.

**Relation To Broader Scientific Literature:**

The contrastive approach followed in the paper is in line with previous works like SAGA, GARField among others that perform 3DGS segmentation using 2D contrastive losses.

**Theoretical Claims:**

N/A

---

> ### Author Rebuttal · Authors · 2025-04-01
>
> Thanks for careful evaluation.
> ## Weaknesses
> > W1: Core part similar to SAGA, GARField ... seem crude and not impactful...
>
> This is a misunderstanding about our method's core. Rather than decomposition, its core lies in the view-aggregated representation, which we believe is not crude.
>
> It consists of two novel modules:
> 1. Cross-view descriptor extraction adaptively captures multi-view semantics for objects.
> 2. Weighted descriptor relevance aggregation refines the representation by assigning weights to descriptors. Unlike hard filtering with information loss, all descriptors are preserved.
>
> LaGa improves 18.7% mIoU over SOTA, and ablation shows +10.6% over baseline. This is clearly impactful.
> > W2: ... require careful tuning ...
>
> The view-aggregated representation needs no sensitive tuning. Its K_max is fixed to 20 across all scenes/datasets. It is stable across K_max from 5–30.
> |K_max|5|10|15|20|30|
> |-|-|-|-|-|-|
> |mIoU|63.4|63.4|64.1|64.0|63.2|
> > W3: HDBSCAN ... how to obtain number of clusters N_p ...
>
> HDBSCAN automatically infers N_p. Its epsilon is fixed to 0.3 without per-scene tuning. As shown below, LaGa has stable performance across a wide range (0-0.3). Too large epsilons lead to object merging.
> |Epsilon|0|0.1|0.2|0.3|0.4|
> |-|-|-|-|-|-|
> |mIoU|62.6|63.0|62.1|64.0|60.6|
> > W4: Sensitive to SAM masks
>
> Boundary regions mis-segmented in one view often appear as interior in others. Through multi-view aggregation, these inconsistencies can be eliminated in 3D, similar to prior work (SemanticNeRF, SA3D).
> > W5: ... reason for k-means ... why not FPS
>
> We adopt k-means to reduce noise by aggregating features locally in the semantic space. In contrast, FPS may preserve more outliers.
>
> For the concern about averaging distant features, we add an denoising to discard features far from their centroids (brings a minimal +0.2% mIoU). The FPS performs worse:
> |FPS|Ours|Ours+Denoise|
> |-|-|-|
> |52.4|64.0|64.2|
> > W6: ... evaluations in 2D ...
>
> No standard 3D benchmark exists for open-vocabulary 3D-GS. We think a sparse set of well-chosen 2D annotated views is a proxy, as neighboring views have redundant information. We check LERF-OVS annotations and find them sufficient in multi-view coverage. This is why most works use it as the main evaluation.
> > W7: Ablation
> > W7-1: ... -DW vs +DW on F. and T.?
> > W7-3: T., Avg pooling ...
>
> The performance fluctuations in F. and T. (Table 4) stem from the sensitivity of fixed k-means clustering. Manually set K may be too small to capture semantic diversity or too large causing outliers. Both are harmful for hard cases.
>
> In Figurines, querying "Pikachu" targets a plastic bag with a Pikachu print. Even K=5 splits the "Pikachu bag" semantics from "bag". By +DW, they are down-weighted as outliers. In Teatime, CLIP sometimes misclassifies a plate as "apple". Avg. pooling or small K merge such descriptors, while K=10 offers better separation. K=20 introduces additional noise. The avg. pooling performs well in Teatime because its objects have more consistent multi-view semantics than other scenes. Avg. pooling is independent of K, simply averaging all multi-view features per object.
>
> Although corner cases are rare, they affect performance (Pikachu: 97% to 31%; Apple: 87% to 21%) and bring instablity. Our adaptive strategy successfully handles them and enables consistent gains with +DW.
> > W7-2: Waldo kitchen...
>
> Thanks for finding this. It is an error caused by swapped DW^c and DW^d results between the F. and W. columns. The correct results are
> ||F.||W.|
> |-|-|-|-|
> |-DW|59.7||62.8|
> |+DW^c|61.6||59.2|
> |+DW^d|60.8||63.9|
> |+DW|64.1||65.6|
> > W7-4: ...adaptive scheme not clear winner
>
> This is a misunderstanding. The adaptive scheme is designed to deliver consistently strong and robust performance across diverse scenes without per-scene manually tuning.
> > W8 (reference): SAGA, GARField ...
>
> They are designed for class-agnostic segmentation, rather than our task. To our knowledge, no prior work evaluates them on LERF-OVS. For comparison, we adapt the released SAGA code for 3D-OVS to deliver the following results:
>
> |F.|T.|R.|W.|Mean|
> |-|-|-|-|-|
> |36.2|19.3|53.1|14.4|30.7|
> ## Questions
> >Q1: Binarized gaussians with 2D GT masks.
>
> During evaluation, for each query, binarized gaussians are projected to 2D via 3D-GS rendering. The resulting 2D masks are compared with GT without any associating.
> >Q2: Cross-View Descriptor Extraction as ablation.
>
> Disabling it is infeasible, as 3D objects require at least one descriptor. Instead, we evaluate it by comparing adaptive scheme with alternatives (Sec. 6.4). '-DW' (Table 4) show results of this module work alone.
> >Q3: Larger clusters L2-norm > 1.
>
> This won't happen since all features are L2-normlized before clustering (L. 192).
> >Q4: Train Langsplat?
>
> No. We do not train LangSplat (neither CLIP features for Gaussians) in LaGa. L632-633 indicates we train three affinity features for each 3D Gaussian to maintain the multi-level SAM segmentation. We will clarify.

---

### Official Review · Reviewer_dCLm · 2025-03-14

**Overall Recommendation:** 3

**Summary:**

This paper proposes LaGa, which explores open-vocabulary 3D Gaussian understanding by decomposing the 3D scene into objects and then establishing view-dependent semantic connections. The proposed approach does not rely on aligning 3D Gaussian semantic features with 2D semantic priors and therefore is simple but effective. Extensive experiments demonstrate the effectiveness of LaGa.

**Claims And Evidence:**

Yes. The superiority of the experimental results is demonstrated through numerical metric comparisons and visual comparisons. Ablation studies and appendix also provide demonstrations on the effectiveness of the proposed method.

**Essential References Not Discussed:**

None.

**Experimental Designs Or Analyses:**

Yes. The authors compare their method with the latest approaches on widely used benchmarks. While there is a gap compared to the latest metrics on 3D-OVS dataset, LaGa achieves the best results on LERF-OVS and ScanNet datasets. The authors also conducted a thorough analysis of the experimental results.

**Methods And Evaluation Criteria:**

Yes.

**Other Comments Or Suggestions:**

typos:

first paragraph of Section 4: “3D objects Section 5.3” -> “3D objects as illustrated in Section 5.3”

**Other Strengths And Weaknesses:**

weaknesses:

1. It would be better to display some failure cases and provide corresponding analysis.

2. I did not find a reasonable understanding of the weighted descriptor relevance aggregation module. The ablation study of average pooling and max pooling does demonstrate the necessity of this module, but there doesn’t seem to be a clear explanation of why this design works. Some further clarification is needed.

**Questions For Authors:**

How do you handle the issue of varying segmentation granularity of SAM across different views? If there are inconsistent segmentations across different views, how does this issue affect the final result? Further discussion on this problem would be beneficial.



######--------------

Update: The rebuttal has addressed my concerns. I will maintain my score of weak accept.

**Relation To Broader Scientific Literature:**

Compared to baseline methods, this paper's contribution lies in addressing view-dependent semantics and establishing cross-view semantic connections to explicitly capture view-dependent 3D semantics.

**Theoretical Claims:**

No proofs or theoretical claims.

---

> ### Author Rebuttal · Authors · 2025-04-01
>
> We sincerely thank you for instructive comments. We hope our response can help clear your concerns.
>
> ## Weaknesses
>
> > W1: ... failure cases and analysis.
>
> Thanks for the suggestion. We summarize key failure cases and provide representative examples:
>
> 1. Bag-of-Words Effect in CLIP:
> When prompted with phrases like 'pirate hat on the rubber duck' or 'cookies in the plate,' CLIP often activates on individual nouns rather than the intended composite concept. This also affects LaGa. Future work may use large language models for better compositional grounding.
>
> 2. Lack of Context in 2D Semantics:
> LaGa extracts 2D object-level semantics by feeding segmented regions into CLIP. While this reduces distraction from unrelated objects, it also removes necessary context. For instance, in Teatime, hooves are unrecognizable without the full sheep, but with the full object, only 'sheep' is detected since missing part-level cues.
>
> These cases reflect the challenges of real-world open-vocabulary perception and the gap between model and human understanding. We will include this discussion and illustrative examples in the paper.
>
> > W2: ... understanding of the weighted descriptor relevance aggregation ... why works needs clarification ...
>
> We hope the following clarification can help address your concern:
>
> Since LaGa does not manually control the segmentation (by SAM) or subsequent semantic feature extraction (by CLIP), the extracted 2D features may contain incorrect or noisy semantics. For example, the spine of a book may look like a 'knife' from certain views. To mitigate this issue, we adaptively weight each semantic descriptor based on two criteria:
> 1. Directional Consistency:
> This metric measures the cosine similarity between each individual descriptor and the global semantics of the object. Descriptors inconsistent with the global semantics receive lower weights. For instance, in the case of the book, descriptors representing a 'passport' align closely with the global semantics 'book,' resulting in higher weights, whereas descriptors resembling a 'knife' are suppressed due to semantic inconsistency.
> 2. Internal Compactness:
> In addition to clearly incorrect semantics, some descriptors result from ambiguous or noisy segmentation (e.g., SAM oversegmentation). They may lack clear meaning yet distort the global feature and compromise directional consistency. To address this, we introduce Internal Compactness, defined as the L2 norm of a descriptor. If the features in the cluster are semantically consistent and have similar directions, their average will have a relatively large L2 norm (close to 1). In contrast, if the features are inconsistent with diverse directions, their vector average will cancel out, resulting in a lower norm. Thus, the norm serves as a confidence measure for semantic reliability.
>
> Together, these criteria help LaGa emphasize descriptors that are both globally consistent and semantically coherent. We will clarify this design further in the paper.
>
> > W3: typos.
>
> Thanks. We will fix them.
>
> ## Questions
>
> > Q1: (1) ... varying segmentation granularity across views? (2) ... inconsistent segmentations affect the final result? Further discussion
>
> Thank you for the insightful suggestion.
>
> (1) To address varying segmentation granularity, we adopt a multi-level modeling strategy (Appendix B). For each 3D Gaussian, we learn three affinity features corresponding to SAM's subpart, part, and whole-level masks, and construct parallel view-aggregated representations. Predictions from all levels are averaged during inference.
>
> (2) When segmentation inconsistency occurs within a level, LaGa resolves it in a statistial learning manner: For a 3D region observed from multiple views with inconsistent segmentation, the dominant label (i.e., the granularity supported by the majority of views) is reinforced through training, enabling convergence toward a stable 3D prediction.
>
> During decomposition, in each level, all masks for the same 3D object are grouped together regardless of granularity. In the cross-view descriptor extraction, masks of different granularities will be assigned to separate semantic descriptors if they exhibit semantic discrepancies. This avoids semantic conflicts during aggregation.
>
> This design also enables **multi-granularity semantic retrieval**: when queried with part-level prompts, the whole-level may exhibit a high response (e.g., 0.5) if it contains the semantic descriptor of that part. At the part level, the queried part will yield a high response (e.g., 0.6), while the remaining regions produce lower responses (e.g., 0). By averaging predictions, the model produces a high response (0.5 + 0.6) for the queried part, a moderately high response (0.5 + 0) for the whole object, and low responses (0 + 0) for unrelated objects. This hierarchical behavior aligns with human cognition and reflects real-world compositional semantics.
>
> We will add the above discussion to our paper to help understanding.

---

### Official Review · Reviewer_ZzT9 · 2025-03-24

**Overall Recommendation:** 3

**Summary:**

The paper introduces LaGa, a novel method designed to improve open-vocabulary 3D scene understanding by tackling view-dependent semantics in 3D Gaussian Splatting. LaGa works by breaking down the scene into 3D objects using multi-view 2D masks. It then aggregates the semantics of these objects through a combination of adaptive clustering and weighted relevance scoring. Experimental results highlight substantial advancements in 3D perception, with LaGa achieving an impressive +18.7% increase in mIoU (mean Intersection over Union) compared to earlier methods when tested on complex datasets.

**Claims And Evidence:**

Yes.
It claims: The view-dependency of 3D semantics causes object understanding to vary with perspective, and simply projecting 2D semantics onto 3D Gaussians results in inaccurate and incomplete scene understanding due to this unaddressed variation.
Evidence:  Figure3 and Figure2.

**Essential References Not Discussed:**

No.

**Experimental Designs Or Analyses:**

I assessed the experimental designs and analyses validating the LaGa method for addressing view-dependent semantics in 3D Gaussian Splatting, focusing on datasets, metrics, ablation studies, and additional analyses. The datasets—LERF-OVS (complex 360° scenes), 3D-OVS (forward-facing, diverse objects), and ScanNet (real-world scans)—are diverse and relevant, testing LaGa across varied scenarios with clear preprocessing steps (e.g., SAM and CLIP usage). Metrics like mIoU and recall rate appropriately measure segmentation accuracy and object retrieval, while ablation studies confirm the value of adaptive clustering and weighting, with strong quantitative support (e.g., 10.6% mIoU drop without clustering).

The designs are mostly sound, with established benchmarks ensuring comparability and analyses like semantic similarity and retrieval integrity (e.g., 50% retrieval failure) reinforcing the problem’s scope. However, potential dataset bias, limited qualitative analysis, an untested recall threshold (0.75), and narrow ablation scope (e.g., omitting decomposition details) are issues that could weaken generalizability and transparency if not addressed. Overall, the experiments are robust but could benefit from broader testing and deeper analysis.

**Methods And Evaluation Criteria:**

The evaluation uses LERF-OVS, 3D-OVS, and ScanNet datasets—covering complex 360° scenes, forward-facing views, and real-world scans—paired with mIoU and recall rate metrics to assess segmentation accuracy and object retrieval completeness. These choices are fitting because they test LaGa across diverse scenarios and directly measure its success in overcoming view-dependency, with supporting analyses and ablation studies reinforcing the method’s design and effectiveness for the problem at hand.

**Other Comments Or Suggestions:**

No.

**Other Strengths And Weaknesses:**

Figure 2 effectively illustrates the motivation behind the study, presenting a clear and intuitive example that guides readers seamlessly through the authors' reasoning.
The subsequent figures and experimental results strongly demonstrate the detrimental impact of this motivation on performance.

**Questions For Authors:**

No for now.

**Relation To Broader Scientific Literature:**

The paper’s introduction of LaGa for open-vocabulary 3D scene understanding via 3D Gaussian Splatting significantly advances prior work like NeRF (Mildenhall et al., 2020) and CLIP (Radford et al., 2021) by explicitly tackling view-dependent semantics—a critical challenge rooted in multi-view stereo where object meaning shifts with perspective; LaGa innovates with 3D scene decomposition and view-aggregated representation, drawing from contrastive learning (NeRF-SOS, Fan et al., 2023), mask-based methods (Garfield, Kim et al., 2024), and clustering, offering a direct 3D solution validated on benchmarks like LERF-OVS and ScanNet, surpassing 2D-reliant approaches like LangSplat (Qin et al., 2024).

**Theoretical Claims:**

The paper lacks formal proofs for its theoretical claims—view-dependent semantics and the efficacy of scene decomposition with semantic aggregation—relying instead on logical reasoning and empirical evidence. I assessed their correctness through coherence, data support (e.g., semantic similarity and retrieval analyses showing view-dependency, and an 18.7% mIoU gain validating the method), and consistency, finding no issues. While not mathematically rigorous, the claims are well-supported by qualitative examples, quantitative results, and ablation studies, making them credible without formal proofs.

---

> ### Author Rebuttal · Authors · 2025-04-01
>
> We sincerely thank you for your time and effort dedicated to evaluating our work. We greatly appreciate your recognition of our motivation, methodological design, and the clarity of our writing. We find the remaining concerns mainly locates in the 'Experimental Designs Or Analyses' section. We hope our responses below will address them.
>
> >  (Experimental Designs Or Analyses) However, potential dataset bias, limited qualitative analysis, an untested recall threshol d (0.75), and narrow ablation scope (e.g., omitting decomposition details) are issues that could weaken generalizability and transparency if not addressed.
>
> **Potential dataset bias:**
> Thank you for the comment. As you mentioned, our evaluation involves diverse benchmarks: LERF-OVS (complex 360° scenes), 3D-OVS (forward-facing views with diverse objects), and ScanNet (real-world scans). We believe this coverage is sufficient to demonstrate the effectiveness of our method. We are not entirely sure which specific bias is being referred to and would be happy to further discuss this concern.
>
> **Limited qualitative analysis:**
> Please kindly refer to our appendix for more visualization results including more visual comparisons, multi-view visualizations, and results on the 3D-OVS dataset. To further strengthen our qualitative analysis, we will incorporate additional visualization results about multi-granularity segmentation cases and representative failure cases to the paper.
>
> **Untested recall threshold (0.75):**
> The threshold of 0.75 was chosen based on analysis of the precision–recall trade-off under different cosine similarity values. Here, precision refers to the proportion of retrieved 3D Gaussians that actually belong to the corresponding 3D object. Low precision indicates that unrelated Gaussians are being retrieved for a given 2D mask.
>
> As shown in the table below, lowering the similarity threshold (e.g., to 0.7) increases the proportion of samples with high recall (>0.9), but at the cost of significantly reduced precision. For example, at 0.7, the average precision of high-recall samples drops to just 13.4%.
>
> From these observations, we find that thresholds in the [0.75, 0.8] range strike a better balance. We conservatively select 0.75, where 50.2% of samples exhibit low recall, clearly demonstrating the challenge of view-dependent semantics. A threshold of 0.8 also yields valid results, with fewer high-recall samples but higher precision.
>
> |Threshold |0.5|0.6|0.7|0.75|0.8|0.9|
> |-|-|-|-|-|-|-|
> |Low / (Low+High) (%)|2.6|13.1|33.7|50.2|65.4|93.8|
> |Average Precision of High (%)|1.7|3.9|13.4|24.1|38.6|48.7|
>
> Note that the average precision does not reach 100%, as CLIP features operate at the semantic level, not the instance level. Therefore, 3D Gaussians with similar semantics but belonging to different objects may also be retrieved, even with a high threshold.
>
>
> **Narrow ablation scope:**
>
> Thanks for the suggestion.
> To further demonstrate the effectiveness of our design choices, we ablate two core components of the decomposition process:
> 1. An ablation study on the **data resampling strategy** used for training the affinity features.
> 2. A **hyperparameter analysis** of the HDBSCAN decomposition algorithm.
>
> Effect of Resampling Strategy:
>
> |              | F.   | T.   | R.   | W.   | Mean |
> |--------------|------|------|------|------|-------|
> | w/o Resampling | 54.8 | 70.5 | 40.8 | 57.5 | 55.9  |
> | w/ Resampling | 64.1 | 70.9 | 55.6 | 65.6 | 64.0  |
>
> As shown above, removing the resampling strategy leads to consistent performance drops across all scenes, highlighting its importance for stable learning of affinity features.
>
> Effect of HDBSCAN Epsilon:
>
> | $\epsilon$ | 0   | 0.1 | 0.2 | 0.3 | 0.4 |
> |--------------|-----|-----|-----|-----|-----|
> | mIoU (%)     | 62.6 | 63.0 | 62.1 | **64.0** | 60.6 |
>
> We empirically set epsilon = 0.3 for all experiments. As the table shows, LaGa achieves strong performance within a broad and reasonable range ($\epsilon \in [0, 0.3]$). Performance degradation only occurs when $\epsilon$ becomes too large, potentially causing unintended merging of semantically distinct objects. This analysis shows that LaGa is not sensitive to this hyperparameter.

---

### Decision · Program_Chairs · 2025-05-01

**Decision:**

Accept (poster)

**Comment:**

The majority of reviewers leaned towards acceptance (scores 4, 3, 3, 3, 1), and the author rebuttal successfully addressed most concerns raised during the review process. One reviewer maintained a strong rejection, mentioning concerns regarding the method's generalizability, the potential need for hand-tuning in the k-means strategy, robustness across different scenarios, and the quantitative comparison with the SAGA baseline. While acknowledging these points were raised, the consensus of the remaining reviewers, the novelty of addressing view-dependent semantics, and the paper's improvements over prior SOTA support its acceptance into the conference. Therefore, I recommend accepting this paper for its contribution to 3D open-vocabulary understanding.

The authors are encouraged to incorporate the feedback from all reviewers and, where possible, further clarify the points regarding tuning sensitivity and the comparison to SAGA in the final version to address the remaining concerns.